# Stable trapping of multiple proteins at physiological conditions using nanoscale chambers with macromolecular gates

Justas Svirelis[1], Zeynep Adali[1,3], Gustav Emilsson [1,3], Jesper Medin [1,3], John Andersson [1], Radhika Vattikunta[1], Mats Hulander[1], Julia Järlebark [1], Krzysztof Kolman [1], Oliver Olsson[1], Yusuke Sakiyama[2], Roderick Y. H. Lim [2] & Andreas Dahlin [1]✉

The possibility to detect and analyze single or few biological molecules is very important for understanding interactions and reaction mechanisms. Ideally, the molecules should be confined to a nanoscale volume so that the observation time by optical methods can be extended. However, it has proven difficult to develop reliable, non-invasive trapping techniques for biomolecules under physiological conditions. Here we present a platform for long-term tether-free (solution phase) trapping of proteins without exposing them to any field gradient forces. We show that a responsive polymer brush can make solid state nanopores switch between a fully open and a fully closed state with respect to proteins, while always allowing the passage of solvent, ions and small molecules. This makes it possible to trap a very high number of proteins (500-1000) inside nanoscale chambers as small as one attoliter, reaching concentrations up to 60 gL$^{-1}$. Our method is fully compatible with parallelization by imaging arrays of nanochambers. Additionally, we show that enzymatic cascade reactions can be performed with multiple native enzymes under full nanoscale confinement and steady supply of reactants. This platform will greatly extend the possibilities to optically analyze interactions involving multiple proteins, such as the dynamics of oligomerization events.

Biomolecules tend to exhibit complex dynamic structures and their interactions are normally reversible with a wide range of lifetimes. To advance our understanding of life on the molecular level, the necessity to analyze single or very few entities is becoming more and more evident as such experiments reveal valuable information that is otherwise lost in an ensemble average[1]. Analysis of single cells or individual nucleic acids[2], including their sequence, is now quite established. The next challenge is to develop methods for analyzing single or small numbers of proteins. This includes both their self-interactions (oligomerization) or binding to other proteins, ligands

etc. Fortunately, great advances in the development of stable fluorescent dyes and controlled conjugation protocols have made optical detection of single proteins feasible[3,4]. Using confocal setups to reduce the excitation/collection volume also enables detection of individual oligomers of medically important proteins[5], thereby revealing heterogeneities in their population.

However, without any confining forces the molecules in the measurement spot can only be observed by fluorescent readout for a few milliseconds before they diffuse away. Because of this severe limitation, much effort has been devoted to the development of

[1]Department of Chemistry and Chemical Engineering, Chalmers University of Technology, 41296 Gothenburg, Sweden. [2]Biozentrum and the Swiss Nanoscience Institute, University of Basel, 4056 Basel, Switzerland. [3]These authors contributed equally: Zeynep Adali, Gustav Emilsson, Jesper Medin. ✉e-mail: adahlin@chalmers.se

trapping methods for small objects, with the aim to extend the observation time beyond the fluorophore lifetime (before photobleaching). Unfortunately, proteins are so small that trapping based on optical[6,7] or electrical[8,9] field gradients becomes difficult, if possible at all, and requires extreme power densities. This influences the sensitive 3D structure of proteins[7] and is thus quite an invasive approach. A crude option is to immobilize molecules on a surface, but this limits their conformational freedom[10] and complicates studies of many processes, such as structure fluctuations in intrinsically disordered proteins[1] and oligomerization. Issues such as these have led to the strong desire for new reliable trapping methods operating in a tether-free manner to avoid surface interactions[11,12]. Very recently, a study with solid state nanopores showed confinement of one or even a few proteins by forcing them into a cavity[13]. Although promising, this method required a continuous voltage to be applied across the nanopore to overcome the forces from Brownian motion, which become strong for objects as small as proteins ($\sim k_{B}T/R$ where $R$ is the radius of the molecule).

An alternative approach to biomolecule trapping is the use of small containers. This provides a much less invasive trapping because no force needs to act on the molecules to confine them. Notably, a hypothetical box with a size of ~100 nm is comparable to cellular compartments and organelles. Given that the walls are repelling, molecules inside such a volume can move around relatively freely, while still appearing static due to the spatial resolution limit of optical detection. In this context, electrostatic traps can be useful for highly charged molecules[14,15] (in particular DNA) but require lowered ionic strength. Alternatively, individual biomolecules can be encapsulated in soft matter constructs such as liposomes, which can then be immobilized on a surface and the molecule inside visualized by total internal reflection illumination[16,17]. However, even the liposome trapping method has clear limitations. First, the yield is not very high, i.e., it is difficult to trap many proteins inside the same vesicle, which limits the possibilities to study interactions. Second, it is not straightforward for small molecules to cross the lipid membrane and access the entrapped protein, which limits studies of, for instance, ligand-induced conformational changes[1]. A technique that does allow access of small molecules while keeping proteins trapped is convex lens induced confinement[18]. The liquid exchange is, however, quite slow as it relies on diffusion over relatively long distances[19]. Furthermore, the chambers do not provide full nanoscale confinement in all three dimensions and the method is typically used to monitor the motion of macromolecules[20].

In this work we present a concept for non-invasive and tether-free trapping using nanoscale chambers in solid state materials functionalized with macromolecular gates. The gates consist of responsive polymer brushes that can be collapsed on-demand by electrical control over the local temperature. For the first time, we show a perfect contrast in diffusive protein transport, i.e. the gates allow proteins to pass unhindered when opened, but also block proteins fully in their closed state. Proteins are captured in the chambers by physisorption to the interior walls while the gates are open, after which the gates are closed, and proteins release from the walls is triggered. The most important advantage of this technology in comparison with existing trapping methods is that very large numbers of proteins can be confined easily in a very small volume (1 attoliter). Furthermore, the trapping time is very long (at least 1 h) and the proteins are confined at physiological conditions without any influence from forces or tethers. In addition, rapid liquid exchange and ligand access is possible while the proteins remain trapped. This is utilized by running enzymatic cascade reactions in solution phase inside the nanochambers, with continuous supply of reactants and removal of products.

## Results

The trapping concept based on nanochambers and macromolecular gates is described in Fig. 1a. A critical component in the system is the responsive polymer brush[21–23] which switches between an extended (closed gate) and a collapsed state (open gate). We have previously shown that hydrophilic polymer brushes can be very strong barriers for proteins[24]. At the same time, the high degree of hydration under good solvent conditions[25] (80% or more) allows small molecules or ions to pass through the gates even when they are closed with respect to proteins, thereby enabling essentially instant liquid exchange in the nanochambers because of the thin brush barrier[24]. We hypothesized that efficient protein trapping could be achieved by letting molecules physisorb to the chamber walls while the gates are kept open. Next, the gates should be closed by letting the polymer chains expand so that they seal the apertures again. Finally, the chemical environment is altered so that the proteins desorb from the chamber walls, leaving them trapped in solution phase. In principle, several candidates can be identified with respect to solid materials and polymer type[21]. In this work, the nanochambers are made in fused silica supports using colloidal self-assembly over large areas[24,26,27] (several $cm^2$), producing either dense arrays (~8 $\mu m^{-2}$, Fig. 1b) or sparse arrays with well-separated chambers (~0.1 $\mu m^{-2}$, Fig. 1c). A semi-transparent 30 nm gold film is introduced on top from which polymers are grafted by material-specific thiol-chemistry[24,26,27]. The opening diameter[27] and the chamber depth[26] are both tunable in the fabrication. We typically used ~100 nm for both these parameters (cross-section image in Fig. 1d), which gives a chamber volume of one attoliter. Furthermore, we use polymer brushes consisting of thermo-responsive poly(N-isopropylacrylamide) (PNIPAM) as gates. PNIPAM brushes can undergo a large reduction in thickness when the temperature is heated above the lower critical solution temperature (LCST) at 32 °C in a physiological buffer[25]. As we have demonstrated for similar nanostructures in air[28], heat can be supplied efficiently and locally to the surface by running a current through the thin gold film[29], applying only a few volts of DC bias. Proteins are adsorbed and released by altering electrostatic interaction forces with silica through the solution pH. This leads to the trapping strategy summarized in Fig. 1e.

A convenient feature of the dense nanochamber arrays is that they enable excitation of surface plasmons in the thin gold film due to the short-range ordered pattern[24,26,27]. The extinction spectrum exhibits a Fano-shaped resonance[30] (a peak and a dip) in the red or near infrared region and this feature shifts upon changes in the local refractive index (RI) on the surface. This label-free readout method, typically referred to as nanoplasmonic sensing, is well established for affinity-based detection[27]. Here we use the intrinsic nanoplasmonic sensor to verify formation and operation of the macromolecular gates. Figure 2a shows the extinction spectrum before and after formation of the initiator layer and the PNIPAM brush on gold by atom transfer radical polymerization (ATRP) with activator regeneration[25]. Considering our previous characterization of the optical properties of the nanostructures[26], the shifts in the spectral peak and dip are in good agreement with the expected thickness for each layer, i.e. ~2 nm for the thiolated initiator[25] and tens of nm for the dry PNIPAM film. Upon resistive heating in water, when the surface temperature crosses the LCST, the polymer collapses into a compact film due to the hydrophobic self-interactions[31]. This is detected by the plasmonic signal (Fig. 2b) because, even though the amount of organic material on the surface remains the same, as the chains collapse they move closer to the metal surface where the sensitivity is higher[25]. As a control, when there is no polymer on the gold, only a small negative signal is seen because of the lowered RI of water at higher temperatures. An electro-thermal calibration using steady-state values is shown in Fig. 2c (power = voltage × current). To estimate the local temperature as a function of electric power we used the value of $10^{-4}$ $K^{-1}$ for the RI change of water[32] (a linear approximation for small changes) and a sensitivity of 147 nm per RI unit for the nanochambers[26]. This results in a defined temperature increment per electric power. (Note that this calibration is not generally applicable as system design comes into

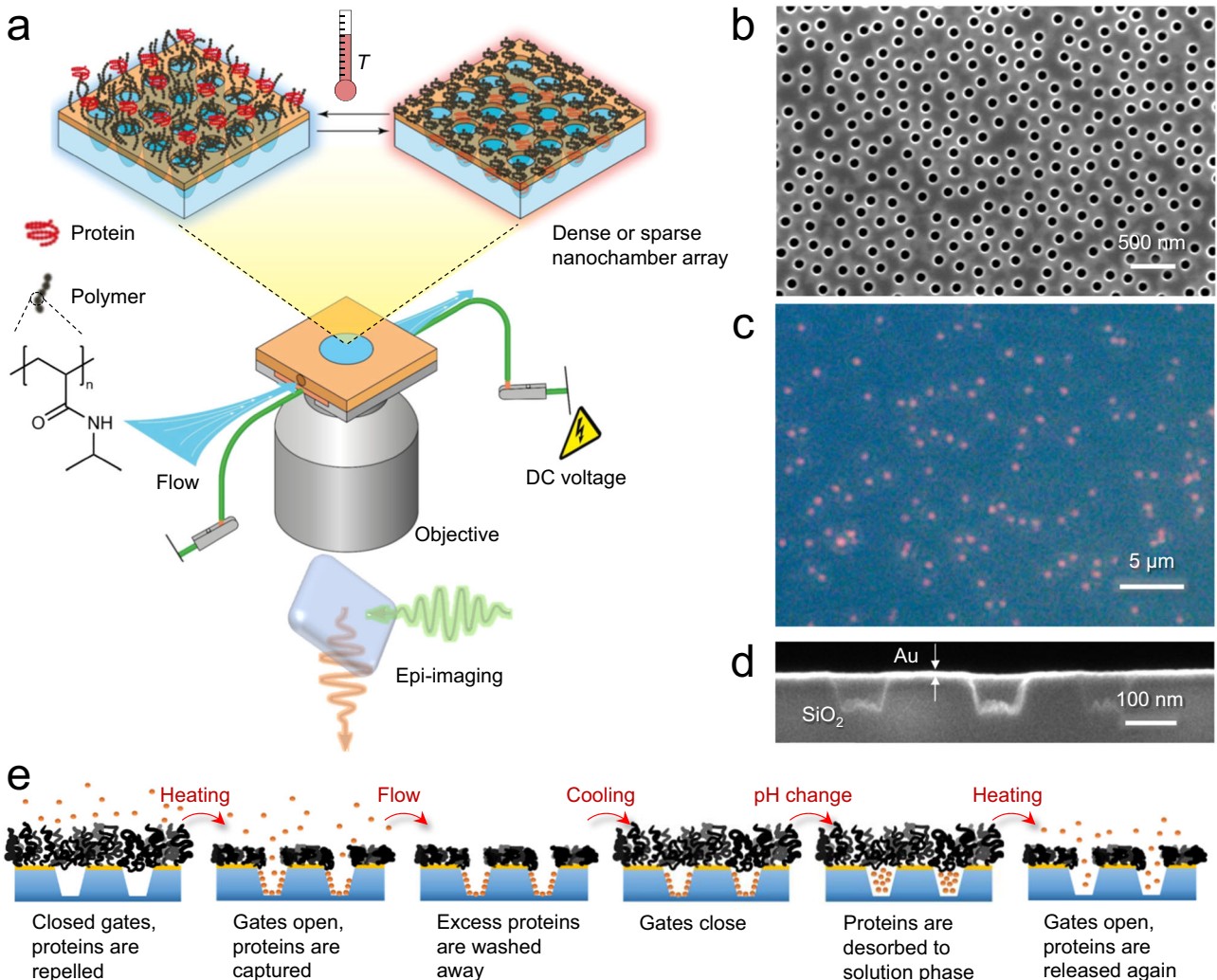

**Fig. 1 | Principle of protein trapping in nanochambers with macromolecular gates. a** Schematic of the system with thermo-responsive polymer brushes on plasmonic nanochambers and resistive heating in a liquid cell. Fluorescence microscopy is performed through the silica substrate. Image partly prepared by Daniel Lara (@danlara on fiverr). **b** Electron microscopy image showing a dense array of nanochambers. **c** Dark field image of a sparse nanochamber array. Each chamber scatters (predominantly red) light. **d** Ion beam cross-section analysis of nanochambers. **e** Trapping process by opening and closing of the macromolecular gates and reversible protein adsorption to the chamber walls. In this work, the gates are operated by heat and the protein physisorption is controlled by pH.

play, including flow cell geometry, convective cooling etc.) The curve confirmed that our results agreed with the expected LCST transition of PNIPAM at 32 °C: we observed that a power of ~1.3 W, corresponding to ~33 °C, was required to maximize the plasmonic signal from the brush collapse. In subsequent experiments we typically used a slightly higher power of 1.5 W (~40 °C). The time trace of the plasmonic signal then consistently showed a small negative signal after the initial positive signal (Fig. 2b), in agreement with some additional heating of water after the PNIPAM brush has fully collapsed.

High speed atomic force microscopy (AFM) in liquid with ultra-sharp tips[24] was used to further investigate the polymer morphology when the gates were opened and closed. Figure 2d shows a plot of the average difference in tip penetration over the pore region compared to the surrounding surface as the temperature gradually increased. Representative frames at room temperature (RT) vs above the LCST are also shown. Interestingly, besides clearly detecting the gate opening, these data also show a sudden increase in penetration depth after about 100 s, indicating a fast transition in the polymer brush morphology. At first sight, this might seem to contradict the plasmonic measurements, where the brush collapse takes a few min (Fig. 2b). However, the plasmonic readout is obtained from a much larger area

(almost 1 cm²) over which there will be significant temperature variations. Most likely, the polymer transition is faster but does not occur at the same time for every location. Indeed, previous work has shown that the collapse can be very fast (10–100 ns) on individual PNIPAM-modified nanoparticles[33]. We believe the transition may be similarly fast also in a pore geometry and propose that each macromolecular gate opens/closes very quickly. Additional liquid-phase AFM data is available in Supplementary Fig. 1. High speed AFM animations in real-time are also available (Supplementary movies 1 and 2).

After verifying gate operation, we investigated the permeability of the gates with respect to proteins for the open and closed states. We have previously shown that poly(ethylene glycol) (PEG) brushes inside nanoscale apertures can be very strong barriers for proteins even in a fully hydrated state[24]. Thus, we expected a sufficiently thick PNIPAM brush to block entry to the chambers at RT, as it is known to be protein repelling on planar surfaces[34]. However, in order to ensure that the gates were open above the LCST, the PNIPAM brush thickness had to be finely tuned (Fig. 3a). If the brush was very thin, the gates were open with respect to proteins both above and below the LCST. If the brush was very thick, the gates were always closed because PNIPAM then spans across the aperture even in its collapsed state[35]. We evaluated

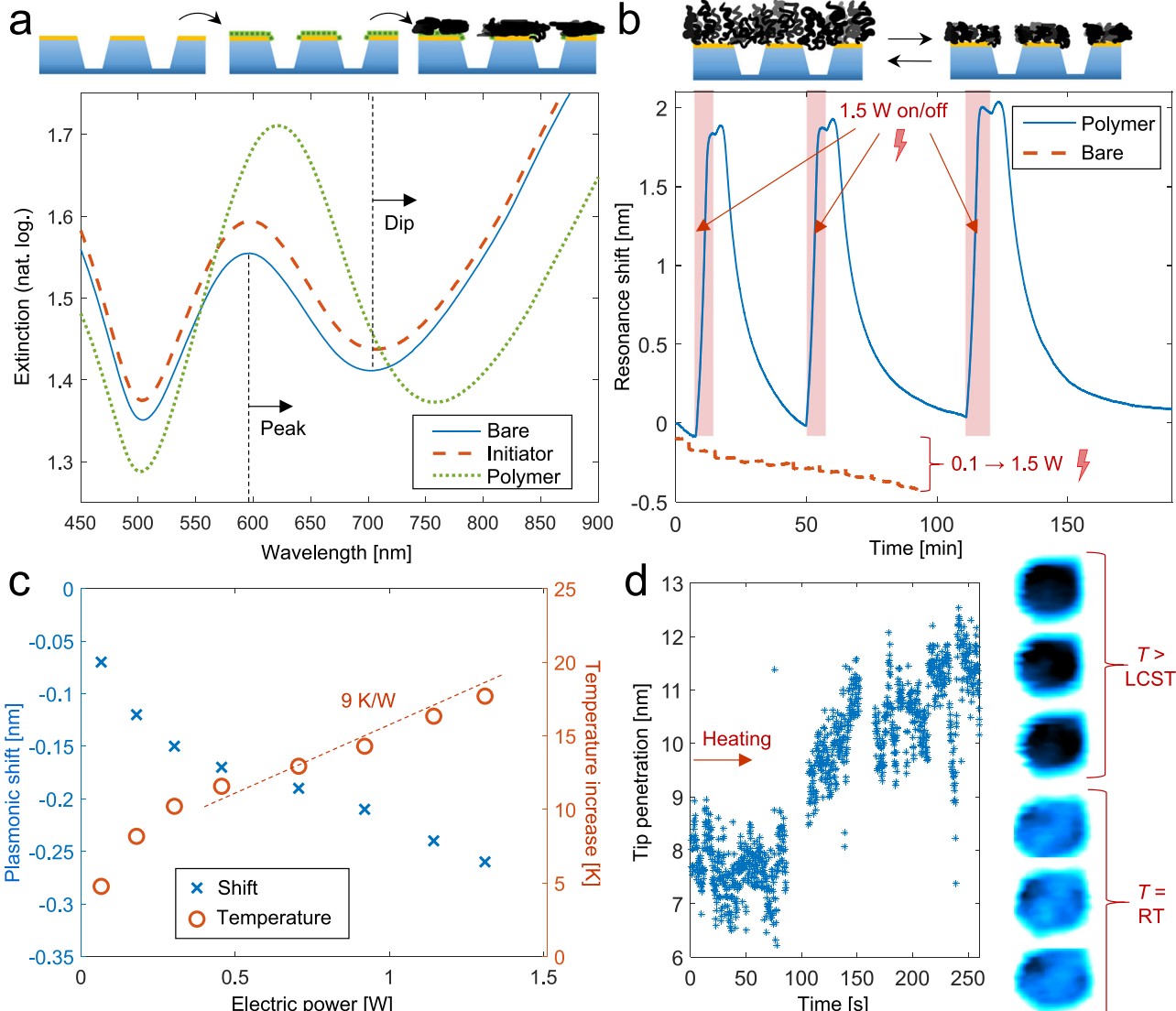

**Fig. 2 | Gate construction and operation. a** Extinction spectra in air showing plasmonic signals from surface initiator binding and polymer brush synthesis. **b** Plasmonic signals monitored in real-time during resistive heating in water by switching the voltage on/off. The resonance shift is caused by the polymer collapsing onto the metal. The control (dashed) shows the response in the absence of a polymer brush when the electric power is increased in steps. **c** Calibration of surface temperature at steady-state heat transport throughout the system using the plasmonic peak shift and RI data of water. The dashed line is a linear fit in the interval between the polymer LCST and 40 °C. **d** High speed atomic microscopy data from a single macromolecular gate. The plot shows the difference in height between the pore opening region and the surrounding area as the system is heated (starting at 0 s). The sudden increase at ~100 s is attributed to the polymer transition. The images show representative frames obtained at room temperature and above the LCST.

different ATRP recipes to control the resulting polymer brush thickness and how it changes in different environments. The thickness in air ($H_{dry}$) was measured by surface plasmon resonance (SPR) and Fresnel models[25]. To obtain the extended brush height in water at RT ($H_{ext}$) and the collapsed height above the LCST ($H_{col}$)[25], PEG in solution was used as a non-interacting probe, which yields a so called exclusion height for both PNIPAM states[25] (Supplementary Fig. 2). As expected[36], we found that one important variable influencing the ATRP kinetics for PNIPAM (and thus $H_{dry}$, $H_{col}$ and $H_{ext}$) was the ratio of methanol/water. Polymerization in pure methanol could provide highly controlled growth without any termination, but the reaction was so slow that it became unpractical (duration >1 day). In contrast, at high water content (70% molar fraction) the polymerization was fast but with many termination events[25]. As a compromise, we found that a water fraction of 60%, where PNIPAM has poor solubility even at RT due to co-nonsolvency[37], resulted in reasonably controlled, fast and reproducible growth (Supplementary Fig. 3). With this recipe, brushes in the thickness range

investigated had a swelling ratio $H_{ext}/H_{dry} = 4.3 \pm 1.3$ and a collapse ratio $H_{ext}/H_{col} = 2.6 \pm 0.6$. The variation was smaller (collapse ratio $2.2 \pm 0.2$) for the planar-surface thickness of interest for gating ~100 nm diameter apertures, which should be $H_{col} < 50$ nm $< H_{ext}$ as we previously showed for apertures modified with PEG[24]. We note that our collapse ratio is slightly lower than previously reported[25], which can be attributed to a higher grafting density with the current ATRP recipe[38].

In order to verify that the macromolecular gates blocked proteins at RT and allowed passage above the LCST, we let bovine serum albumin (BSA) adsorb to the silica surface inside the nanochambers. This was achieved by lowering the pH to 5-6, thereby making the protein less negatively charged (pI 4.5-5) which reduces the electrostatic repulsion from silica, an effect which is well known[39]. Complementary experiments with SPR and quartz crystal microbalance (QCM) were used to verify that BSA adsorbs irreversibly to silica under these conditions, i.e. it is not spontaneously desorbing, but can be released again by raising the pH to 8-9 (Supplementary Fig. 4). Neither

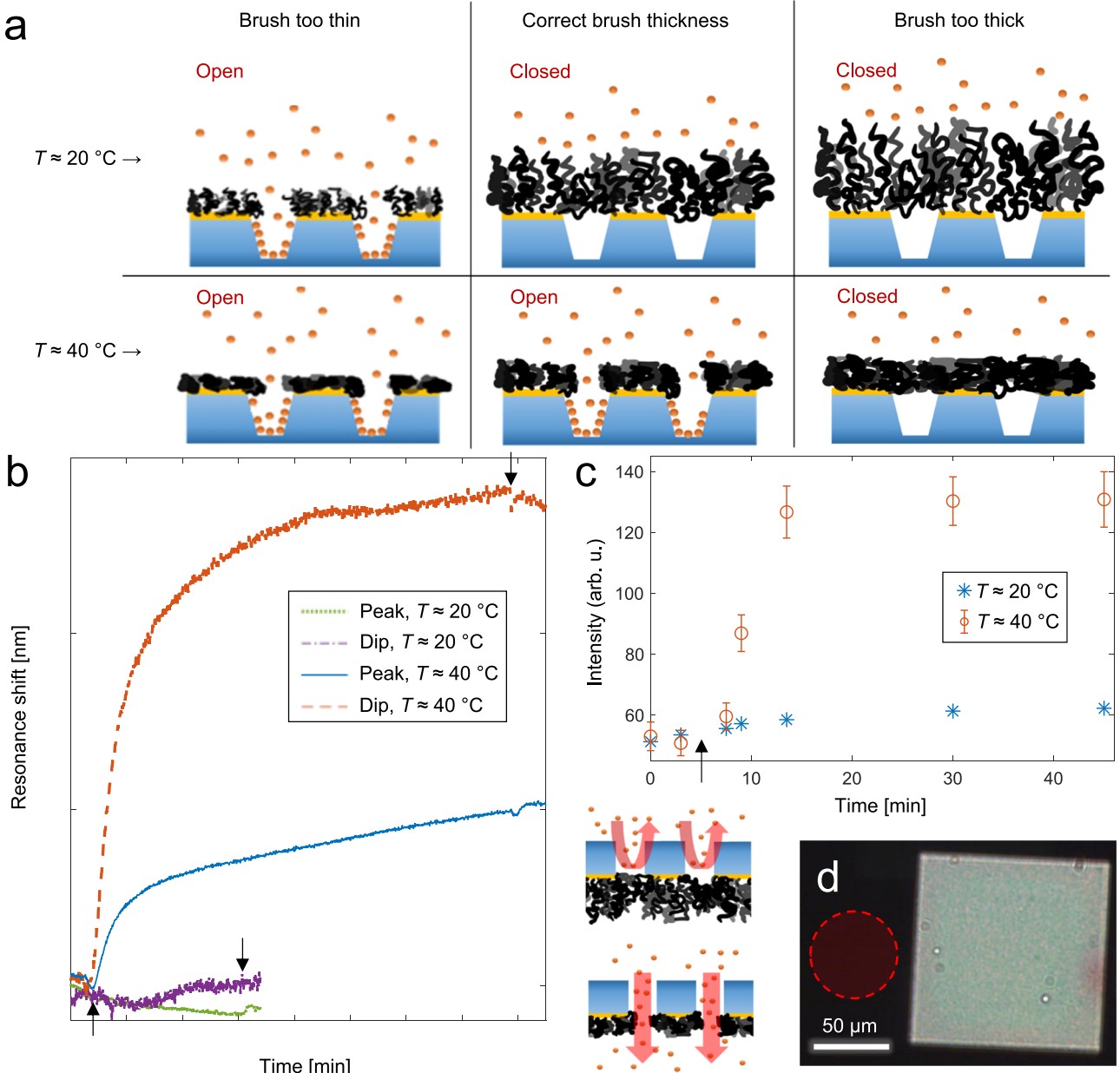

**Fig. 3 | Verifying open and closed states of the macromolecular gates.**
**a** Illustration showing that the brush thickness must close the apertures (with respect to proteins) at RT, but keep them open (allow protein transport) above the LCST. Too thin brushes give gates that are open in both states. Too thick brushes give gates that are closed in both states. **b** Plasmonic signals from BSA (50 μgmL$^{-1}$) adsorption inside nanochambers (at pH 6.0) for brushes with the correct thickness. No signals are observed when BSA is introduced at room temperature. Arrows indicate injection and rinsing. **c** Thermally gated ultrathin silicon nitride membrane with polymer-modified pores. Fluorescent BSA (introduced at 5 min) diffuses through the membrane in the heated state only. Error bars are equal to two standard deviations and were obtained by measuring intensities from different locations around the membrane. **d** Optical microscopy image showing the membrane with pores as observed in transmission mode. The red circle illustrates a typical region where the fluorescence intensity was measured.

the heating to ~40 °C[40] nor the pH changes[41] alter the structure of the protein. Figure 3b shows that for the PNIPAM-modified nanochambers at pH 6, no plasmonic signal was observed from BSA at RT (experimental uncertainty ~0.1 nm over ~1 h), confirming that the hydrated brush acts as a strong barrier. In contrast, when the electric heating was on, protein adsorption was detected. Control experiments were done on dense nanochamber arrays with a thin repelling PEG coating instead of PNIPAM, showing similar binding kinetics and saturation signals (Supplementary Fig. 5). This shows that the PNIPAM nanochambers are fully open above the LCST, i.e., the diffusive protein transport is not significantly hindered. Note that the plasmonic signal originates predominantly from BSA adsorption to silica inside the

nanochambers and not onto the collapsed PNIPAM brush on gold. This is in agreement with previous work[25] and supported by additional control experiments on planar surfaces (Supplementary Fig. 6). The large signal in the extinction dip as compared to the peak further proves localized binding inside the nanowells[26,42]. Note that this holds for BSA: other proteins may adsorb more on the collapsed polymer brush. However, if they do they will be released again when the brush hydrates again at RT[29], which means that it is irrelevant for our trapping strategy.

To further benchmark the contrast in the temperature-controlled protein transport, we also looked at fluorescently labeled BSA introduced on one side of a membrane with pores having the same

geometry as the nanochambers but with the bottom connecting to another reservoir[24] (Fig. 3c). The fluorescence was measured on the other side, next to the membrane (Fig. 3d), so that any intensity increase is due to molecules that have passed through the membrane to the other side. At RT, no significant fluorescence increase was seen, while the proteins clearly diffused through the membrane when heated. This illustrates that transport through nanopores can be thermally gated by grafted PNIPAM, a topic that has been investigated previously, but only with much thicker membranes such as track-etched polycarbonate[35,43–48], anodized alumina[49] or mesoporous silicon/silica[50,51]. In such work the permeability has mostly been investigated with ion currents[46,48] or pressure driven flow of only the solvent itself[52]. Other studies have looked at passive transport of small molecules[49–51], DNA[43], or synthetic polymers[44,47] through PNIPAM-modified pores under different conditions. However, when it comes to proteins, transport seems to have been investigated only under pressure driven flow through track-etched membranes[44,45]. We emphasize that the strong contrast in open vs closed states which we present here has never previously been demonstrated, i.e. previous systems have been leaky in their closed states. We attribute this mainly to the low variation in aperture diameter and shape for our nanostructures[27] in comparison with previous porous membranes.

After verifying the operation of the electro-thermal macro-molecular gates, we attempted to realize trapping of multiple proteins according to the strategy in Fig. 1e. The nanoplasmonic signal during a whole trapping and release experiment is shown in Fig. 4a. The protein blocking ability, the gate opening, and the protein adsorption is first monitored as described previously (Figs. 2b and 3b), thereby confirming that the PNIPAM brush has a suitable thickness (see results for other thicknesses in Supplementary Fig 7). Protein adsorption is followed by gate closure by cooling to RT, which does not lead to baseline recovery, as expected when BSA remains adsorbed to the walls inside the nanochambers. To release the BSA into solution phase, the pH was then increased to 8.0. The pH can be quickly exchanged inside the nanochambers when PNIPAM is in its hydrated state because a brush (in contrast to other constructs such as lipid bilayers) allows passage of solvent, ions and even small molecules[24]. When the pH was increased, a minor decrease in the plasmonic signal was consistently observed (Fig. 4a) with kinetics similar to those of BSA desorption (Supplementary Figs. 5 and 7). We attribute this mainly to the inhomogeneous sensitivity distribution associated with the plasmonic near-field[53] rather than proteins leaking from the chambers (Supplementary Note 1). Regardless, the signal stabilized at a value much higher than the baseline, showing that proteins remain inside the nanochambers, as expected since the gates remain closed (PNIPAM is not pH-responsive). To further verify that the trapping was achieved with proteins in solution phase, we again used the electric current to heat the surface above the LCST. The heating response then looked different, and the initial baseline was recovered after cooling (Fig. 4a). This is consistent with proteins leaving the nanochambers as they open again, confirming that they were indeed not attached to the surface.

To further prove the tether-free trapping of proteins we performed fluorescence measurements on the nanochambers following the same procedure as in Fig. 4a but with labeled BSA. The adsorption/desorption behavior on silica was not influenced by the fluorophores (Supplementary Fig. 8). Figure 4b shows the fluorescence intensities measured in epi-mode through the glass support. The intensity from a control region without any nanochambers was subtracted (Fig. 4c). Note that the total area of exposed gold is the same on both regions because the area of the walls of each aperture in the gold film matches the cross-section area. After exposure to BSA at RT and pH 6, a small fluorescence signal was observed from the nanochambers compared to the gold film. We attribute this to a minor amount of protein adsorption to the PNIPAM modified gold and a higher transmission of fluorescent light through the nanochamber region of the surface

(Supplementary Note 2). Regardless, the fluorescence from the nanochambers increased much more after BSA had been introduced while the system was heated above the LCST (Fig. 4b). Importantly, after closing the gates and raising the pH to 8 to make the proteins desorb, this intensity remained for at least 1 h. Furthermore, the intensity went down as expected when the gates were opened again by electric heating, allowing the proteins to escape. Thus, the stable capturing of multiple proteins in solution phase observed in the nanoplasmonic measurements was independently confirmed by fluorescence readout.

Nanochambers should be addressed individually to facilitate studies of the entrapped molecules inside them. This is also a requirement for resolving heterogeneities in the nanochambers themselves, i.e., to examine how many that trap proteins as intended. To investigate this, we used sparse arrays and correlated the positions of nanochambers observed in dark field imaging with fluorescence images. Figure 5a shows an example of dark field and fluorescence images after trapping labeled BSA in solution phase, i.e. after the pH increase step. The red circles, which clearly dominate in number, mark nanochambers for which there was also a significant increase in fluorescence intensity at the same position. The blue circles show nanochambers from which no fluorescence could be detected and the yellow circles indicate spots from which fluorescence was detected without any nanochamber present. The nanochambers were clearly visible in dark field with a homogenous scattering intensity as expected[54] and thus we consider it highly unlikely that there are undetected nanochambers at any of the yellow circles. Hence, the yellow circles most likely originate from protein adsorption to the planar gold due to defects in the nanostructure and/or the polymer brush. Fortunately, this has no effect when using the platform for analysis of proteins: fluorescent signals from positions where there is no nanochamber should simply be excluded.

Further statistical analysis of individual nanochambers after each step in the trapping process (example data in Supplementary Fig. 9) revealed that the majority captured proteins as intended. After analyzing hundreds of nanochambers, we found that 80–90% contained proteins after completing the process for trapping them in solution phase. This is a conservative estimate since we cannot exclude that there are undetected fluorescent proteins in some nanochambers. A small fraction of malfunctioning chambers can be expected if there is a lack of lateral uniformity in the PNIPAM brush. Even if the average thickness of the macromolecular gates is ideal (Fig. 3a), it may still vary over the sample surface, in which case some chambers will not function as intended because $H_{ext}$ and/or $H_{col}$ are not correct (Supplementary Note 3). Note that it does not matter if the PNIPAM brush is a bit too thick or a bit too thin: both cases lead to the blue circles exemplified in Fig. 5a. Overall, we consider our yield of working nanochambers good for an initial study and since our method is compatible with massive parallelization by imaging (each sample has tens of millions of nanochambers), one can simply exclude those that do not show any trapped proteins.

The fluorescence intensities from trapped proteins in Fig. 5a were quite faint compared to the background, which is largely because the wide field microscopy system is not optimal for detecting low intensities. In addition, gold causes relatively high autofluorescence[55]. In this pioneering study, we worked with gold because of its plasmonic activity, which has helped us develop the macromolecular gates and verify their operation, as shown above. To prove that higher signal to noise is feasible, we performed some proof-of-concept tests where gold was replaced with palladium[55] and confocal laser scanning was used to further increase the ratio of signal vs background intensities (Fig. 5b). Using 2D scanning in the plane of the nanochambers, the intensity from pixels correlating with a nanochamber position was up to 50 times higher than the background, even at very high scan rates (15 µs per pixel). Note that Pd can be chemically modified by thiol chemistry just like Au[56].

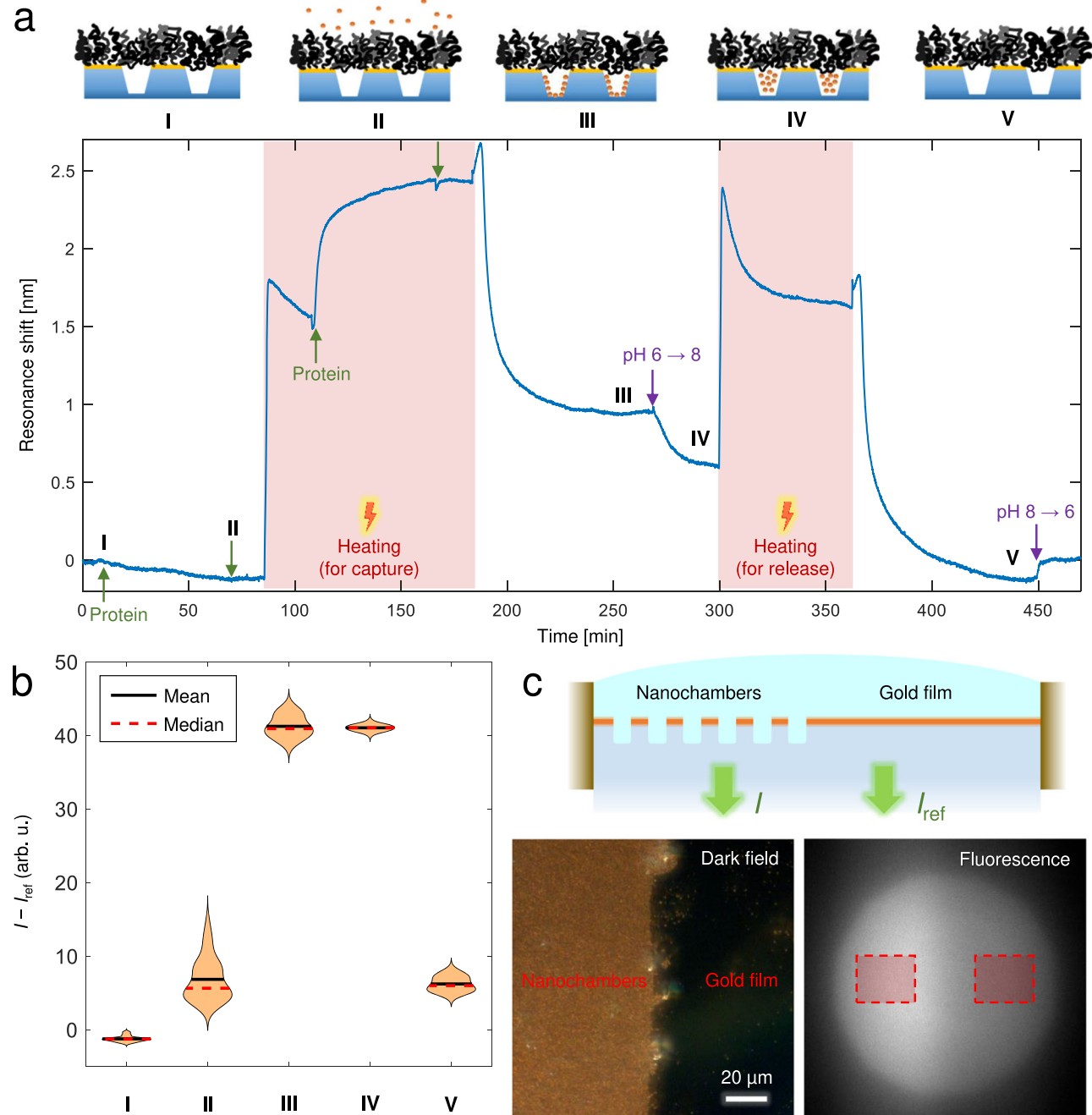

**Fig. 4 | Trapping proteins inside nanochambers. a** Nanoplasmonic signal (extinction dip shift) during an entire trapping and release experiment. The Roman numerals indicate the different states of the system as illustrated by the top schematics. The arrows indicate injections and rinsing. The baseline (zero shift) is recovered after returning to the initial temperature and pH (within the uncertainty ±0.1 nm). **b** Fluorescence intensity difference between dense nanochambers and planar gold after the different steps in the trapping process (I-V as in panel **a**). For the experiment shown, 30 intensity values were used for each violin plot and ~1 h elapsed between each data set (during which heating/cooling and rinsing occurred). All intensities were measured at RT and without proteins in the bulk solution above. **c** Dark field and fluorescence imaging showing the nanochambers and the control zone with planar gold. The fluorescence image illustrates the two regions used to collect average intensities.

An important parameter for any trapping technique is the time during which biomolecules remain captured. In our system, a detailed investigation of the trapping time is complicated because of drifts/uncertainty and gradual photobleaching of fluorophores (which was observed when using high intensities). Still, we can conclude that the trapping time is at least ~1 h, which is clearly orders of magnitude higher than for many existing tether-free methods such as recently reported electrokinetic[11] or optical[6] traps. In particular, the nanochambers trap proteins much longer than the

characteristic time until photobleaching when analyzing single molecules[3,4] (typically ~1 min under laser illumination). Naturally, a short trapping time may be enough in certain applications when it is sufficient to get a fluorescent snapshot of molecules diffusing by[5,55]. However, biomolecular dynamics often occur over long timescales (hours or more)[1], in which case an extended trapping time is necessary to probe single binding/unbinding events. Yet, the truly unique advantage with our method is the large number of proteins that can be confined to a zone smaller than the diffraction limit.

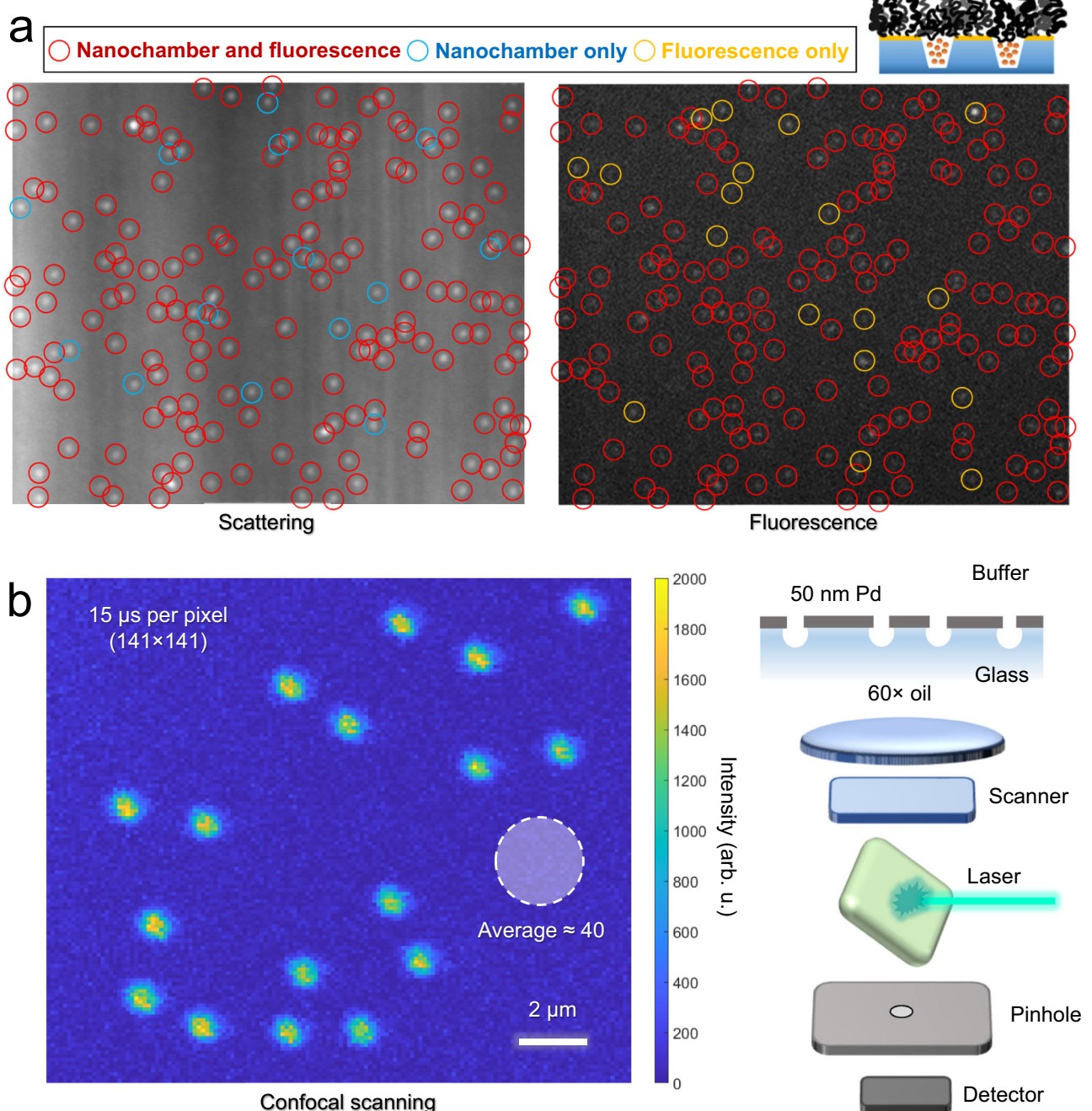

**Fig. 5 | Analysis of individual nanochambers (sparse arrays). a** Example of nanochambers visualized by light scattering in dark field and the corresponding fluorescence image taken after trapping proteins tether-free. In each experiment, all nanochambers in a chosen region (at least 100) were analyzed. The pattern of red circles is the same in both images, showing nanochambers that correlate with spots in the fluorescence image. Blue circles show nanochambers that lack a significant fluorescence signal. Yellow circles in the fluorescence image do not coincide with a nanochamber. The imaged region is approximately 50 μm in width. **b** Confocal laser scanning 2D image of nanochambers with Au replaced by Pd, showing improved signal (vs background) from proteins for an exposure time of 15 μs in each pixel.

To estimate the number of BSA molecules inside each chamber, we note that the silica surface area is 31,400 nm$^2$. The saturated coverage of BSA on silica may be as high as 500 ngcm$^{-2}$[39], but we use the more conservative recent estimate of ~200 ngcm$^{-2}$ for adsorption at pH 6, based on Kubiak-Ossowska et al.[41] This translates to almost 600 proteins and a concentration inside each nanochamber of over 60 gL$^{-1}$. Notably, this is orders of magnitude higher than existing container-based tether-free trapping methods, which rely on random encapsulation during preparation[10,17]. In principle, the concentration can be increased further by reducing the nanochamber diameter, thereby increasing the surface to volume ratio, though this will also make it increasingly challenging to get the right PNI-PAM brush thickness. Note that here we have investigated the upper limit of how many proteins that can be trapped. Naturally, it is also possible to trap a smaller number of proteins by simply interrupting the adsorption before saturation in the capture phase. Alternatively, to capture single proteins, one can instead use conditions where adsorption does not occur and tune the concentration so it is comparable to one molecule in one attoliter (~1 μM). Many nanochambers will then not contain any molecule, but this is

compensated by the massive parallelization, i.e. only data from those chambers that show a fluorescent signal would be analyzed.

An important feature of our nanochambers with macromolecular gates is the possibility to quickly introduce small molecules to the trapped proteins. Based on our previous work with PEG brushes[24,57], we expected that the highly hydrated PNIPAM brush would be permeable to small molecules (-1 kgmol$^{-1}$) at RT. However, previous literature is inconsistent with respect to this point. While several studies show that hydrated PNIPAM brushes are permeable to small molecules at RT[44,45,50], others suggest that this is not the case[49,51]. To resolve this issue, at least for our system, we introduced amine groups on the interior of the nanochambers by silanization[58] and exposed them to a small (752 gmol$^{-1}$) dye which forms a covalent bond with amines. Clear fluorescence was observed from the nanochambers after exposing them to the dye (Supplementary Fig. 10), which confirms that it was able to move through the brush and bind to silica. The silanization results also illustrate that the nanochamber interior can be chemically modified. For instance, during trapping experiments it can be positively charged instead of negative, which is suitable for adsorbing anionic proteins. Also, although BSA does not undergo any structural changes by its pH-reversible adsorption-desorption process on silica[40], this may not be the case for all proteins, which makes the possibility to perform silanization important (Supplementary Note 4).

As a final result, we show an example of how the unique advantages of the gated nanochamber trapping platform can be utilized by running an enzymatic cascade reaction with continuous supply of reactants. This illustrates the two main advantages offered by our system: that several proteins can be trapped together in solution-phase and that small molecules easily can access them. This differs strongly from compartmentalization in microemulsions whose volume is a billion times larger[59,60]. Nanoporous materials can be more densely loaded with enzymes and offer interesting ways to enhance cascade reactions[61], but this relies on adsorption to the surface of the material. We also note that various fusion-protein capsids have been presented for specific cascade reactions and can be densely loaded with enzymes[62–64]. However, the enzymes are then part of the inner shell and not in solution phase, as is desired[65]. Furthermore, all these constructs are designed to maximize catalytic activity in bulk systems, not as a platform to analyze enzyme activity by fluorescence methods on the single compartment level.

To generate a cascade reaction inside the nanochambers, two different enzymes (β-galactosidase, GAL, and glucose oxidase, GOX) were captured in a simple manner using off the shelf nanochamber chips placed in a pH 6 buffer containing a mixture (50 μgmL$^{-1}$ of each enzyme) heated to 35–40 °C. The solution was then cooled down to RT after which the sample was rinsed with water and dried. The plasmonic signal could once again be used to confirm protein loading (Fig. 6a). We verified that both GAL and GOX could be adsorbed and released from silica by changing the pH, just like BSA (Supplementary Fig. 11). By conjugating different dyes to each enzyme type (fluorescein or Cy3), we could also confirm that they could be trapped together in the chambers by dual channel fluorescence (Fig. 6b).

The enzymatic cascade reaction was initiated simply by introducing a buffer containing lactose, horseradish peroxidase (HRP) and Amplex red on top of the surface. GAL and GOX break down the lactose substrate inside the nanochambers. This produces H$_2$O$_2$, which diffuses out through the PNIPAM brush to meet HRP, which in turn oxidizes Amplex red to generate the fluorescent derivate resorufin (Fig. 6c). Note that HRP is excluded from entering the nanochambers and thus the cascade reaction illustrates both substrate access and product removal through the brush barrier. The fluorescence increased over time when the compounds were introduced, confirming the generation of resorufin. However, care must be taken in this kind of measurements since Amplex red can form resorufin even without HRP and/or H$_2$O$_2$. This effect led to high variation in the

experimental results, but the autooxidation could be accounted for by measuring the fluorescence under identical conditions in the absence of lactose. The higher intensity in the presence of lactose confirmed the cascade reaction inside the nanochambers both in terms of kinetics (Fig. 6d) and end values (Fig. 6e).

The successful reaction cascade clearly shows that the brush barrier solves the well-known problem of how to achieve continuous supply of reactants/products to/from confined enzymes[65]. Using alcohol dehydrogenase (ADH), we also managed to transport the redox cofactor nicotinamide (NADH) back and forth through the brush barrier (Supplementary Fig. 12). This, in turn, made it possible to perform localized enzymatic reactions coupled through cofactor cycling with GOX[59] (Supplementary Fig. 13), where HRP and ferricyanide were added to oxidize NADH[66–68] (Supplementary Fig. 14). The array of nanochambers with trapped enzymes can be looked at as a biocatalytic surface that performs a certain reaction, except that the enzymes can be in solution phase, thereby avoiding the risk of reduced activity due to surface immobilization[69]. Additionally, catalytic enhancement effects due to proximity or substrate channeling[70] can be studied without immobilization scaffolds[71] and with the enzymes in their unmodified native state.

## Discussion

We have presented a method for trapping of biomolecules, with focus on proteins, for very long observation times and at physiological conditions. The nanoscale compartments with high concentrations of different proteins can be thought of as mimics of native cell environments. While trapped, the proteins are in aqueous solution and not exposed to any external forces, nor are they tethered to a surface. This is achieved by lithography-defined nanostructures functionalized with thermo-responsive polymer brushes. Our method offers the unique advantage of capturing several hundreds of proteins at once inside a nanoscale compartment as small as one attoliter, while still being compatible with efficient liquid exchange through the polymer brush barrier. In principle, single proteins (enzymes or others) can also be captured and their conformational dynamics studied over extended time by Förster resonance energy transfer (FRET). In this case, the main advantage over existing platforms[17] would be the ease of introducing small ligands through the hydrated brush barrier. Furthermore, the optical readout by fluorescence from individual nanochambers is compatible with massive parallelization. We believe our method also has an additional advantage: the trapping is very simple to perform once the polymer-functionalized nanochambers are prepared. Thus, it can become a widely used platform for analyzing single/few protein molecules by state-of-the-art fluorescence techniques. Since the chips could be stored for later use (no aging effects were observed during several months), one can envision polymer-functionalized nanostructures distributed to other users for use in existing microscopes. The local resistive heating of the surface is convenient for analyzing the operation of the macromolecular gates, but for trapping the proteins we have shown that it is sufficient to place the sample in a temperature-controlled solution.

Using the gated nanochambers, several studies relevant for molecular biology can be envisioned that are not possible with any other technology. As an example, the dynamics of protein oligomerization can be monitored by FRET. The number of captured proteins in the nanochambers is similar or slightly higher than the number of units found in oligomers of medically important proteins such as α-synuclein[5]. Furthermore, the effective protein concentration inside the chambers is at least an order of magnitude higher than that used in bulk experiments of fibril formation[5]. This means that the dynamics of the critical nucleation step, which has only been possible to investigate by simulations[72], should be possible to study by FRET by introducing different dyes to the proteins in a controllable manner[5]. The extended observation time will then make it possible to investigate aspects such

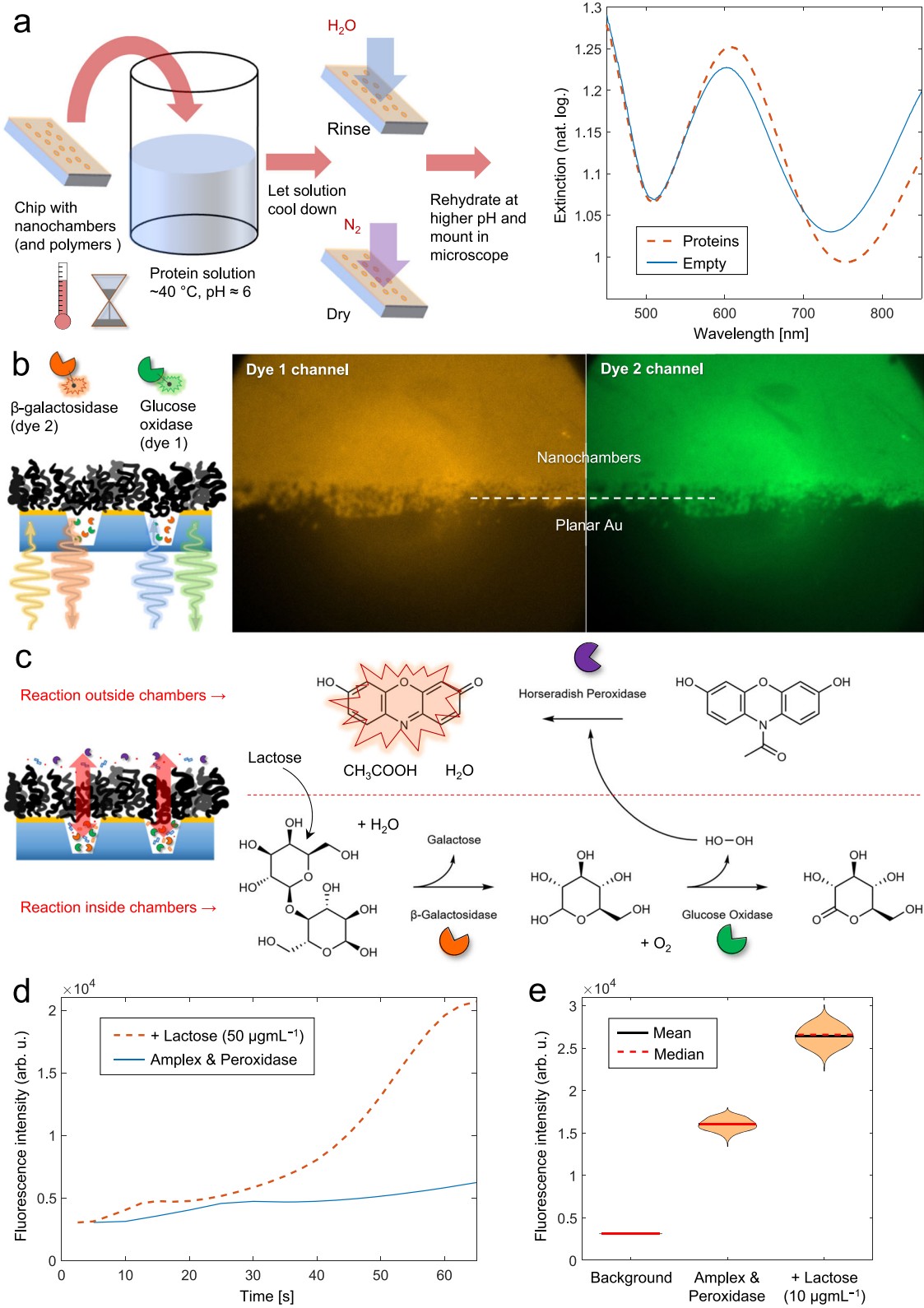

**Fig. 6 | Continuous enzymatic cascade reactions in nanoscale reaction chambers. a** Simple preparation procedure using a nanochamber chip. The extinction spectra in air confirm protein loading through the spectral shift. **b** Dual-channel fluorescence imaging (pseudo colors added) of GAL and GOX labeled with different dyes and trapped together in nanochambers. **c** Cascade reaction scheme. Inside the nanochambers, GAL and GOX (here unlabeled) convert lactose (which has diffused in) to $H_2O_2$. Outside the nanochambers, HRP creates a fluorescent compound by

oxidation from $H_2O_2$ (which has diffused out). **d** Fluorescence intensity increase from nanochambers with trapped native (unlabeled) GAL and GOX upon introducing lactose, HRP and Amplex red in the solution above (at zero seconds). The control intensity trace had no lactose included. **e** Violin plots from an experiment where intensities were measured ~15 min after introducing the reactants to a nanochamber chip. For each data set, 63 values were obtained by resampling. Note that for the background, the variation is too small to be visible.

as kinetics and reversibility of the oligomerization process. In addition, it is possible to introduce small molecules to the captured proteins and investigate how this influences their interactions, which shows relevance also for drug development studies.

## Methods

### Chemicals

All chemicals used were purchased from Sigma-Aldrich unless stated otherwise. Water was ASTM research grade Type 1 ultrafiltered water (MQ, 18.2 MΩcm). $H_2O_2$ (30%) and $NH_4OH$ (28–30%) were from ACROS or ThermoFischer Scientific, while $H_2SO_4$ (98%) and ethanol (99.5%) were from SOLVECO. The initiator for ATRP was α-bromobutyrate-11-undecanethiol (Prochimia). Chemicals used for polymer synthesis were $N, N, N', N'', N''$-pentamethyldiethylenetriamine (PMDTA), $N$-iso-propylacrylamide (NIPAM), $CuBr_2$ and ascorbic acid. Thiolated PEG (2 kgmol$^{-1}$) was from Laysan Bio. Amplex UltraRed was from Thermo-Fischer. Dyes functionalized with an NHS ester group (for conjugation) were from Lumiprobe (Sulpho-Cy3) and Thermo Scientific (fluorescein).

The proteins used were: Native BSA ≥ 98%, heat shock fraction, protease free, fatty acid free, essentially globulin free (Sigma-Aldrich product A7030, Lot SLBX0893). BSA conjugated with Alexa488, 1:7 molar ratio (Thermo Fischer catalog number A13100, Lot 2418516). GAL from *Aspergillus oryzae* (Sigma-Aldrich product G5160). GOX Type VII from *Aspergillus niger* (Sigma-Aldrich product G2133). ADH from *Saccharomyces cerevisiae* (Sigma-Aldrich product A3263). HRP Type II (Sigma-Aldrich product P8250).

Unless stated otherwise, the running buffers were either (1×) phosphate buffered saline (PBS; 10 mM phosphate, 137 mM NaCl and 2.7 mM KCl) for pH below neutral or (1×) borate buffered saline (BBS; 10 mM $Na_2B_4O_5(OH)_4$ and 150 mM NaCl) for pH above neutral. The pH was adjusted with 3 M HCl or 1 M NaOH solutions and was controlled within a ± 0.2 interval.

### Nanostructure fabrication

The nanochambers were prepared by colloidal self-assembly (140 ± 5 nm polystyrene-sulfate, Microparticles Gmbh) as described previously[26]. Plasma etching was used to tune the entrance diameter ($O_2$ to etch the colloids) and the chamber depth ($CF_4 + O_2$ to etch silica). Gold (30 nm) was deposited by electron beam heated physical vapor deposition (Lesker PVD 225), preceded by a ~1 nm Cr adhesion layer and followed by a 20 nm $Al_2O_3$ protection layer[27]. Nanochambers with Pd (50 nm) instead of Au were made by HF etch of the glass support, which was 0.17 mm ±5 μm borosilicate. Nanopores in 50 nm silicon nitride membranes were prepared by electron beam lithography using a negative resist (ma-N 2403) spin-coated on the pre-made membranes, followed by development (ma-D 525)[30]. Au and $Al_2O_3$ was then deposited as for the nanochambers and lift-off was performed (mr-Rem 700). The pores were created by dry etching silicon nitride using the same protocol as for $SiO_2$. Alternatively, colloidal lithography was performed on the membranes instead of electron beam lithography, which creates identical pores but in a short-range ordered pattern[30] (as for the dense nanochamber arrays).

### Surface preparation

Nanochambers were cleaned with RCA1: 1:1:5 volume ratio of $NH_4OH$, $H_2O_2$ and water for 20 min at 75 °C, rinsed with MQ water, sonicated for 5 min in EtOH (99.5%) and dried under flow of $N_2$. SPR chips were cleaned with piranha solution: $H_2SO_4$ (95–97%) and $H_2O_2$ mixed in ratio of 3:1 for 20 min. Take note! Piranha solution heats spontaneously and foams vigorously when in contact with organic material. Afterwards, the surfaces were rinsed with MQ water and dried with $N_2$. Silica coated QCM and SPR chips were cleaned with UV $O_3$ (placed under a 90 W mercury vapor lamp for 10 min), sodium dodecyl sulfate (2% solution for 30 min) rinsed with MQ water and dried with $N_2$. Then another UV

$O_3$ step was performed for 30 min, followed by rinsing in ethanol and drying with $N_2$.

Gold SPR sensors (50 nm Au) were purchased from the SPR instrument manufacturer. Silica coated chips were created by performing an additional atomic layer deposition step (Oxford FlexAL) aiming for a thickness of 15 nm. The gold was first cleaned by $O_2$ plasma for 1 min.

Assembly of the initiator layer was achieved by placing samples in 1 mM thiol initiator solution in ethanol overnight (at least 18 h). After self-assembly, samples are sonicated (35 kHz) in ethanol for 1 min and dried with $N_2$. In some experiments a diazonium salt was used to create an initiator layer instead. Ascorbic acid was then used to create an aryl bond to the metal[73]. We noted no difference in the properties of the polymer brushes depending on the type of chemical bond that grafted PNIPAM to the surface, while the ATRP protocol (see below) clearly influenced properties such as growth kinetics and swelling properties.

### Surface-initiated polymerization

ATRP with activator regeneration by electron transfer was used to synthesize PNIPAM brushes on functionalized gold surfaces with the following concentrations of reagents: 0.48 M NIPAM, 0.0064 M PMDETA, 0.01 M ascorbic acid and 0.0006 M $CuBr_2$. The solvent was a binary mixture of MQ water and methanol (99.8% anhydrous). The final volume of the polymerization solution was 49 mL in a glass jar with a screw cap. The methanol was further dried with 0.3 nm molecular sieves and filtered with 0.2 μm syringe filter before use. Appropriate amounts of NIPAM, PMDETA and $CuBr_2$ were dissolved and the solution was degassed with $N_2$ at a flow rate of 560 mLmin$^{-1}$ for 30 min. 5 min before degassing was finished, the solvent mixture with the reagents was transferred with a transfer needle and a pump to a degassed jar, where the sample surfaces were placed on a teflon rack. The reaction was initiated by injecting ascorbic acid (previously dissolved in MQ water) into the jar and run at room temperature with stirring at 500 rpm. The volume above the reaction solution was purged with $N_2$ during the polymerization. The reaction was stopped by opening the jar cap and transferring the sample rack to a beaker with ethanol. Afterwards, sample surfaces were dried thoroughly with $N_2$ and stored in plastic holders for later use.

### Extinction spectroscopy

The extinction spectra were measured in a custom setup with a fiber coupled array spectrometer and lamp (B&WTek) with collimating lenses[27]. Extinction was defined as the natural logarithm of the ratio of incident and transmitted light intensities. The same custom-made flow cell was used as in the wide-field fluorescence microscopy experiments. In some experiments where quantitative information was extracted, a linear baseline correction was performed if the same drift was observed every time the system was idle throughout the measurement.

### Wide-field epi-fluorescence microscopy

Fluorescence experiments were performed using an inverted Axio Observer optical microscope equipped with an Andor IXon Life CCD and an Axiocam color camera, LED light sources and a 50× objective (air, NA = 0.55, WD = 9.0 mm) in epi-mode. A peristaltic pump (Ismatek) was used to direct flow through the flow cell. For detection of Alexa Fluor 488, an excitation beam splitter that reflects and transmits light with 452–486 nm and 500–528 nm wavelengths, respectively, was used together with an emission filter transmitting 501–527 nm. The illumination was at 475 nm. For fluorescein, Cy3 and resorufin, standard settings in the software were used. Dyes were conjugated to enzymes by mixing 0.5 moles of dye per 1 U (activity unit) of enzyme at a concentration of 1 UmL$^{-1}$. Violin plots were made using the default kernel density estimation in Matlab.

Fluorescence images were obtained at predefined time points with the Andor camera. Background intensities were measured with the same setup parameters and illumination but before introducing any fluorescent compounds. Acquired images were converted to TIFF files with ImageJ and values were extracted in Matlab. Complementary dark field images were obtained either with Axiocam or Andor cameras using the same objective, but the illumination was performed with a halogen lamp.

## Confocal microscopy

Nanochambers were prepared on No. 1.5H cover slips and images were acquired using an oil immersion objective (60×, N.A 1.4) on a Nikon Ti-E A1+ at a laser wavelength of 490 nm and emission bandpass of 510–560 nm with the pinhole set to 0.4 AU. The detector was a photomultiplier tube.

## Resistive heating

To generate a local temperature increase during the extinction spectroscopy or fluorescence experiments, resistive heating was used with isolated copper wires, conductively glued to the edges of the sample surface. A chosen DC voltage was applied with a Gamry Interface 1000 potentiostat.

## Silanization

Amine groups were introduced on the silica surface after ATRP using aminopropylsilatrane[58]. The compound was introduced at 460 μM in EtOH for 10 min, followed by rinsing in EtOH and annealing at 70 °C for 1 h.

## Surface plasmon resonance

Angular spectra of SPR sensors were measured using a Bionavis SPR Navi 220A instrument. All data shown was acquired with the 670 nm laser diode. Measurements were performed at 25 °C unless stated otherwise and the flow rate was 30 μLmin⁻¹. Gold sensor spectra (with initiator layer) in air were measured beforehand as reference in the Fresnel models. The non-interacting probe method was used to determine PNIPAM exclusion heights in the extended and collapsed states[25], using PEG as probe. The running buffer was PBS (pH = 7.4). PNIPAM collapsed brush heights were determined in the same way but the spectra were recorded at 35 °C.

## Quartz crystal microbalance

Measurements were performed using a Q-Sense E4 instrument (Biolin Scientific) equipped with a peristaltic pump (Ismatec). Standard gold crystals were purchased from QuartzPro and silica coated sensor crystals were purchased from Biolin Scientific. The system was rinsed with buffer until a stable baseline was achieved. Data is shown at the 3rd overtone.

## Atomic force microscopy

High-speed AFM was performed as described previously[24]. Drift correction was performed to keep the pore centered. The temperature was increased by heating the whole measurement setup. A thermometer was used to verify that the solution has crossed the LCST of PNIPAM.

Contact mode images were obtained in water using an NTEGRA AFM (NT-MDT) with Tap300AI-G tips (BudgetSensors). In Gwyddion (http://gwyddion.net/) mean plane subtraction was used to level the data, as well as row alignment with a 2nd degree polynomial.

## Data availability

The data supporting the key findings of this study are available within the article and the Supplementary Information. Additional raw data are available from the corresponding author upon request. Source data are provided with this paper.

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

## Acknowledgements

This work was financed by the European Research Council (grant agreement 101001854) and the Erling-Persson Family Foundation (Starting Grant 2017), both awarded to A.D.

## Author contributions

J.S. performed the majority of the experiments. Z.A. performed initial experiments testing macromolecular gates. G.E. developed polymerization protocols and polymer thickness determinations. J.M. performed measurements on enzymes. J.A., R.V. & J.J. fabricated and characterized nanostructures. R.V & M.H. performed confocal microscopy. O.O. helped setting up controlled resistive heating of the samples. K.K., Y.S. & R.Y.H.L. performed AFM measurements. A.D. conceptualized the idea, led the project and wrote the manuscript (with input from all authors). Z.A., G.E and J.M. contributed equally.

## Funding

## Competing interests

The authors declare no competing interests.
