## [Peer Review File · Nature Communications]

Stable Trapping of Multiple Proteins at Physiological Conditions Using Nanoscale Chambers with Macromolecular GatesREVIEWER COMMENTS

Reviewer #1 (Remarks to the Author):

Svirelis et al. reported the fabrication of nanoscale chambers for trapping protein molecules using macromolecular gates. The work is carefully executed and the manuscript clearly written. The opening sentence in the Abstract gave an impression that the authors were going to develop single-molecule sensing technique, but actually it was not. They were going the other direction of trapping proteins as many as possible. Other than that, the flow of the paper is smooth and comfortable to read.

The idea of trapping proteins in a nanocavity is not new, therefore efficiency and functionality are the key parameters of the work. I have the following questions after I read the manuscript.

1. On page 11, the authors discussed the polymer gate response time. There are three pieces of information and taken together it is quite confusing. First, the authors quoted that the polymer transition can be extremely fast (10-100 ns) on individual PNIPAM-modified nanoparticles. Second, the authors wrote "At first sight, this might seem to contradict the plasmonic measurements, where the brush collapse takes a few min (Figure 2B)". Third, in Figure 2D, High speed atomic force microscopy showed nanochamber opening in about 100-120 s. A clearer explanation is desperately needed here.

2. In Figure 3, the authors emphasize the importance of the right thickness of the brushes. "The brush thickness must be higher than the radius of the aperture at RT, but smaller above the LCST. Too thin brushes give gates that are open in both states. Too thick brushes give gates that are closed in both states." But they did not show how they measure the aperture at RT and above the LCST. Further, they did not show how to control the exact thickness of the polymer brush.

3. In Figure 4a, "the baseline is recovered after returning to the initial temperature and pH". The recovery is quite evident but what does the baseline mean. There are clear differences

on the resonance shift.

4. On page 22, the authors discussed the trapping time and trapping concentration in nanochambers. They claim that the trapping time is at least 1h which is much longer than other methods. It occurs to me that a overly long trapping time should be considered as a disadvantage. Why is it good here? For the trapping density, they estimated the value according to other literature. Is there a direct way of measuring the protein density using fluorescent proteins? Although silica was used in both cases, the overall constructs might be very different. Therefore, it would be better to have a direct evidence for the density estimation.

Reviewer #2 (Remarks to the Author):

Key Results: This paper outlines an effective approach for the entrapment of protein molecules at very high concentration inside nanoscale enclosures/chambers with a volume of 1 attolitre. The proteins are free to move in solution, i.e. they are not tethered to the inner walls of the enclosures – this is advantageous since any restriction of conformational freedom will yield observations that are not truly representative of the proteins' behaviour in vivo. The system works by allowing the enzymes to enter the enclosures and then trapping them inside by exploiting a polymer brush which acts as a macromolecular gate at the chamber entry point. It can be opened and closed by adjusting the temperature, which facilitates the entry and subsequent entrapment of the proteins inside. The chambers are composed of silica and a layer of gold on top provides a surface to which the polymer brushes can be chemically attached; note that the gold does not cover the chamber aperture – it surrounds it. The polymer brush gate is poly(N-isopropylacrylamide) and when closed, it is permeable to small molecules and ions such as enzyme substrates. The authors tested their approach with a model enzyme cascade and proved entry of substrate and exit of product.

General Comment: I am not experienced in the techniques used throughout this paper (SPR and QCM)- my expertise, relating to this paper, is the behaviour of enzyme systems under nanoconfined and highly concentrated conditions. However, I did find the paper very clearly

written and relatively easy to follow. Some of the comments below are most likely borne out of a lack of familiarity with the techniques used, but since the potential readers of this paper may be coming from the viewpoint of an interest in enzyme confinement, and the power that studying enzymes under these conditions affords, then I think it might be worthwhile to address these comments and to explain certain things a bit more- things that the authors no doubt feel are obvious.

Validity: The experiments presented are robust and appropriate controls have been carried out.

Significance: I think this work is highly significant in the field of studying enzymes functioning under confinement, however, please see my comments below (numbers 2 and 11) referring to the citation of other notable works that have achieved what the authors claim is unique (note, this word should be used with caution) to their approach – i.e. managing to entrap proteins in tiny enclosures at very high concentrations and where they are untethered and function as they most likely would in vivo. The authors also need to expand on other ways to study single or very few proteins in small enclosures with the addition of relevant citations in this field and more discussion on how their own approach could be exploited for this.

Data and Methodology: As stated above, I am a non-expert in the techniques SPR and QCM however, I have gone through the data in detail and have raised any questions below in my numbered list of comments. The experiments and controls are robust.

Analytical approach: The analyses are sound with potential artefacts ruled out or reasoned out. In Figure 4B the error bars represent variation from subsequent acquisitions – which I presume means that it was the same experiment just measured a few times – these error bars therefore represent the error of the instrument rather than a true experimental replicate. Also, have the experiments presented throughout the paper been replicated? Especially figure 4A which is the main experiment proving the technique and also figure 6, its application. I assume many such experiments were carried out during the course of this work, so the authors could mention this in the figure legends or in the SI somewhere.

Suggested Improvements: See the list of my comments below. In addition to these, the authors could try another enzyme cascade for further proof of concept. The cascade they have used is an excellent choice since it is well characterised in literature, and it nicely shows the entry of substrate and the exit of product. Another idea might be to use an enzyme or enzyme cascade that uses a cofactor such as nicotinamide, the reduced form of which, absorbs at 340 nm and could be easily detected outside the chamber. This concept could be extended if an enzyme pair was used that recycled the cofactor back and forth between each other- enzyme 1 could be trapped in the chamber and enzyme 2 outside – recycling the cofactor back and forth through the permeable gateway would be a beautiful result.

In their conclusions, the authors suggest that their approach will enable the study of oligomerisation at high protein concentrations by FRET- but at the start of the paper in the introduction, they claim that its power lies in the potential to be used for analysing “single or small numbers of proteins” – again, as I have outlined for the abstract, I find this confusing- please clarify that it is both. Apart from oligomerisation, the authors should list the other possible ways in which their approach could be exploited since they write “...several studies relevant for molecular biology can be envisioned...”

One claim is that it could be used to study single protein molecules – what behaviour in a single protein molecule could be monitored and how? Would it be possible to detect enzyme motions in one molecule somehow?

Clarity and Context and References: Please see comments regarding addition of extra references. Also, the paper mostly discusses the use of the approach to study proteins at very high concentrations but also mentions its potential for studying single protein molecules- the authors need to include more citations and discussion on the latter. And also, be clear in the distinction.

Comments

1. The abstract opens with a sentence stating that the ability to detect and analyze single or few biological molecules has proven very important for understanding their function and interaction mechanisms, then a few sentences later the authors point out that it has been more difficult to entrap biomolecules under “physiological conditions”- I think the author

needs to be more specific in what they refer to here by “physiological conditions” which I believe they mean to be high local concentrations of biomolecules entrapped inside cellular compartments. Then at the end of the abstract the authors claim that their approach will extend the possibility of single/few molecules analysis – I found this all to be muddled: first saying that this has already been tackled in literature then saying but it hasn’t under physiological (high local concs and entrapped) but then back to single/few molecules. The abstract needs to be clarified in this respect. Are the authors interested in achieving highly concentrated/crowded conditions under nanoconfinement or are they interested in single/few entities being nanoconfined? Or both? I think it is both.

2. The 3 statements in the bullet points below are quite sweeping – I think it is important that the authors cite more work in this field and have suggested citations below which show other work in which entrapment of proteins at high local concentration inside compartments has been achieved as has the ability for small molecules to cross the barrier. These references should be included.

- ♣ Abstract: reaching concentrations up to 60 g/L, which is orders of magnitude higher than for existing container-based technologies.

- ♣ Page 4: However, even the liposome trapping method has clear limitations. First, the yield is not very high, i.e., it is difficult to trap many proteins inside the same vesicle, which limits the possibilities to study interactions. Second, it is not straightforward for small molecules to cross the lipid membrane and access the entrapped protein, which limits studies of, for instance, ligand-induced conformational changes.

- ♣ Page 5: The most important advantages of this technology in comparison with all existing trapping methods is that very large numbers of proteins can be confined very easily in an extremely small volume

The authors should point out other examples where entrapment at a high yield and also the ability for small molecules to cross the barrier has in fact been achieved, for example, this work by Bai and coworkers involving protein cages doi.org/10.1021/acscatal.1c02221 which entrapped two enzymes at 0.89 and 5 mM in a compartment with 65 nm diameter which was permeable to NADP(H).

And also this work DOI: 10.1126/science.aaz6802 by Erb and coworkers involving the encapsulation of a 16-enzyme crotonyl-CoA carboxylase/reductase cascade

Other work which could be cited includes this work by Cornelissen and coworkers DOI: 10.1039/c0sc00407c; Douglas and coworkers doi.org/10.1021/cb4006529;

Perhaps MOFS and COFS should also be discussed in the context of this work – for example, even the much-tested model system that the authors have used (glucose oxidase (GOx) and horseradish peroxidase(HRP)) as their proof of concept, has also been used in a MOF doi.org/10.1002/anie.201710418 this work might be worth discussing as a comparison.

3. Page 7- a reference is needed for this sentence – “...gold film is introduced on top from which polymers are grafted by material-specific chemistry” what is the material-specific chemistry?

4. Page 10 Figure 2B: I found this figure confusing because at first glance, I mistakenly assumed that the two heating events (just after 0 min and at ~50 min) were aligned with the illustration above – in other words I thought the PNIPAM layer remained thick until the third heating event after 100 minutes (the red panel for this one being aligned to the illustration above in which the PNIPAM layer had collapsed). I may be being pedantic here, but I think that the authors should consider this accidental alignment.

Also, explain the figure in more detail for the readers who are interested in enzyme entrapment but who are non-experts in plasmon resonance data.

5. Figure 4D: could temperature be added as another X axis at the top? This would show the precise temperature at ~100 s.

6. Figure S4A: the background signal is ~50 Hz and then BSA is injected which causes the signal to decrease to below 40 Hz then rinsing occurs at ~50 min after which, the signal remains unchanged. At 70 min I presume the system is cooled because the pale red shaded area returns to white (write heating above the red area as was done for figure 4A)- at this point the signal decreases further to ~ -18 Hz so, is this indicative of maximum BSA adsorption? I presume it is. Then when the pH is changed to 8 the signal increases which must be indicative of BSA desorbing- why then, does the signal not return to the original

baseline if all the BSA has desorbed?

7. Figure S5: this is a control using PEG on gold which the authors state is protein-repelling- the maximum peak resonance shift is 3.5 (dip) and ~1.8 (peak)- the authors state that these values are not very different to those obtained in figure 3B which are 1.4 (dip) and 0.3 (peak). Are these considered similar for this technique?

8. Page 16: the description of the experiment in Figure 4A: I recommend a more detailed description of the time course in 4A as it is the main experiment which proves the technique. For instance, there is a second injection of BSA just after 100 secs. Why was this required since BSA had already been injected at the start of the experiment? Was the BSA that had been injected first, washed away? Or did it remain in the surrounding solution ready to enter the pores? I assumed the latter but if it is the former then add more detail to the figure legend.

9. Figure S6: QCM data which is presented as frequency shift and dissipation change – For convenience, the authors should present the data as weight change too- this would have helped with understanding of figure S4A also.

10. Supplementary note 4: Last two sentences – suggestion of imidazole as a way to desorb the enzymes would not be ideal since the solution in the chamber would then have a very high ionic strength which would not be considered ideally physiological – a selling point of this work.

11. Page 22 This sentence is sweeping, “....60 g/L. Notably, this is orders of magnitude higher than existing container-based trapping methods.” The authors should cite the work I have listed in point number 2 above which show other entrapment methods that achieve this – another notable approach that should be cited, because it achieved extremely high concentrations of proteins trapped in a porous electrode which is also accessible to small molecules and ions, can be found here DOI:10.1021/acs.chemrev.2c00397 or here <https://doi.org/10.1073/pnas.221412312>

Reviewer #1:

Svirelis et al. reported the fabrication of nanoscale chambers for trapping protein molecules using macromolecular gates. The work is carefully executed and the manuscript clearly written. The opening sentence in the Abstract gave an impression that the authors were going to develop single-molecule sensing technique, but actually it was not. They were going the other direction of trapping proteins as many as possible. Other than that, the flow of the paper is smooth and comfortable to read.

We thank the reviewer for the feedback. We have revised the abstract to clarify that although our platform certainly can be used for single molecule studies, it is probably more valuable for studying multiple proteins and their interactions.

The idea of trapping proteins in a nanocavity is not new, therefore efficiency and functionality are the key parameters of the work. I have the following questions after I read the manuscript.

1. On page 11, the authors discussed the polymer gate response time. There are three pieces of information and taken together it is quite confusing. First, the authors quoted that the polymer transition can be extremely fast (10-100 ns) on individual PNIPAM-modified nanoparticles. Second, the authors wrote "At first sight, this might seem to contradict the plasmonic measurements, where the brush collapse takes a few min (Figure 2B)". Third, in Figure 2D, High speed atomic force microscopy showed nanochamber opening in about 100-120 s. A clearer explanation is desperately needed here.

The polymer transition is faster than what can be measured by the plasmonic signal due to temperature variations across the surface. We have revised the main text and the figure caption.

2. In Figure 3, the authors emphasize the importance of the right thickness of the brushes. "The brush thickness must be higher than the radius of the aperture at RT, but smaller above the LCST. Too thin brushes give gates that are open in both states. Too thick brushes give gates that are closed in both states." But they did not show how they measure the aperture at RT and above the LCST. Further, they did not show how to control the exact thickness of the polymer brush.

We define a closed pore based on the inability of proteins to translocate. Based on our previous work with another polymer, the pores close approximately when the brush extension on a planar surface becomes comparable to the radius. This has now been clarified.

We control the thickness through the ATRP recipe by altering solvent composition, concentration of reactants and, of course, the time of the reaction. A minor clarification has been added.

3. In Figure 4a, "the baseline is recovered after returning to the initial temperature and pH". The recovery is quite evident but what does the baseline mean. There are clear differences on the resonance shift.

The baseline is the zero-peak shift, in other words the beginning of the experiment. The signal eventually returns to this value (within the experimental uncertainty). We have clarified the caption.

4. On page 22, the authors discussed the trapping time and trapping concentration in nanochambers. They claim that the trapping time is at least 1h which is much longer than other methods. It occurs to me that a overly long trapping time should be considered as a disadvantage. Why is it good here? For the trapping density, they estimated the value according to other literature. Is there a direct way of measuring the protein density using fluorescent proteins? Although silica was used in both cases, the overall constructs might be very different. Therefore, it would be better to have a direct evidence for the density estimation.

This is a good point. For some studies, where it is sufficient to obtain a "snapshot" of a species, it can be advantageous to have a short trapping time so that the event frequency increases. This would be the case in, for example, zero mode waveguides. However, many biomolecular dynamics occur over much longer timescales and an extended trapping time is then necessary to study them on the single/few molecule level. For instance, for reversible interactions, this becomes evident from typical association and dissociation rate constants. We have added a brief discussion where the trapping time is mentioned.

Quantitative interpretation of fluorescence intensities is generally very difficult, which is why we went for another approach. We assume that all proteins that were adsorbed on silica in each chamber become trapped and utilize the known surface area inside each chamber. The surface coverage of BSA on silica is known to be in the range of a few hundred ng/cm² and it adsorbs in a monolayer until saturation (see cited papers). Surface curvature should not play a major role considering the protein size. There is certainly an error margin in our estimate, but it is quite clear that the number of proteins inside each nanochamber must be on the order of a few hundred, which represents a much higher density than other techniques for tether-free trapping.

Reviewer #2:

Key Results: This paper outlines an effective approach for the entrapment of protein molecules at very high concentration inside nanoscale enclosures/chambers with a volume of 1 attolitre. The proteins are free to move in solution, i.e. they are not tethered to the inner walls of the enclosures – this is advantageous since any restriction of conformational freedom will yield observations that are not truly representative of the proteins' behaviour in vivo. The system works by allowing the enzymes to enter the enclosures and then trapping them inside by exploiting a polymer brush which acts as a macromolecular gate at the chamber entry point. It can be opened and closed by adjusting the temperature, which facilitates the entry and subsequent entrapment of the proteins inside. The chambers are composed of silica and a layer of gold on top provides a surface to which the polymer brushes can be chemically attached; note that the gold does not cover the chamber aperture – it surrounds it. The polymer brush gate is poly(N-isopropylacrylamide) and when closed, it is permeable to small molecules and ions such as enzyme substrates. The authors tested their approach with a model enzyme cascade and proved entry of substrate and exit of product.

We thank the reviewer for the feedback. The summary of the key results is very accurate.

General Comment: I am not experienced in the techniques used throughout this paper (SPR and QCM)- my expertise, relating to this paper, is the behaviour of enzyme systems under nanoconfined and highly concentrated conditions. However, I did find the paper very clearly written and relatively easy to follow. Some of the comments below are most likely borne out of a lack of familiarity with the techniques used, but since the potential readers of this paper may be coming from the viewpoint of an interest in enzyme confinement, and the power that studying enzymes under these conditions affords, then I think it might be worthwhile to address these comments and to explain certain things a bit more- things that the authors no doubt feel are obvious.

We agree and made clarifications accordingly as described below.

Validity: The experiments presented are robust and appropriate controls have been carried out.

Significance: I think this work is highly significant in the field of studying enzymes functioning under confinement, however, please see my comments below (numbers 2 and 11) referring to the citation of other notable works that have achieved what the authors claim is unique (note, this word should be used with caution) to their approach – i.e. managing to entrap proteins in tiny enclosures at very high concentrations and where they are untethered and

function as they most likely would in vivo. The authors also need to expand on other ways to study single or very few proteins in small enclosures with the addition of relevant citations in this field and more discussion on how their own approach could be exploited for this.

The reviewer is obviously very knowledgeable about methods for enzyme confinement. We have included the references suggested below and explained how our system differs from previous work, both here and in the manuscript. In brief, the main points are the high density of tether-free enzymes in our chambers and the ease of access/removal for substrates/products through the brush barrier.

However, we also point out that in the end our platform is intended for studies of proteins in general, not only enzymes, by confining them below the optical diffraction limit. The studies that the reviewer refers to are certainly relevant, but often aim to maximize catalytic activity of certain enzymatic reactions, not to study proteins by fluorescence methods.

Data and Methodology: As stated above, I am a non-expert in the techniques SPR and QCM however, I have gone through the data in detail and have raised any questions below in my numbered list of comments. The experiments and controls are robust.

Analytical approach: The analyses are sound with potential artefacts ruled out or reasoned out. In Figure 4B the error bars represent variation from subsequent acquisitions – which I presume means that it was the same experiment just measured a few times – these error bars therefore represent the error of the instrument rather than a true experimental replicate. Also, have the experiments presented throughout the paper been replicated? Especially figure 4A which is the main experiment proving the technique and also figure 6, its application. I assume many such experiments were carried out during the course of this work, so the authors could mention this in the figure legends or in the SI somewhere.

All experiments have been repeated several times to confirm the results. It is correct that the error bars (now violin plots) in figure 4B represent the measurement error in that particular experiment. This has been clarified in the caption. Absolute intensity values would not be accurate to compare between experiments as they vary too much. However, additional data to confirm the results can be provided. For figure 4B, an example is given in Figure R1 below, showing values from another experiment performed in the same way. (Data from at least two additional similar experiments is also available.) Plasmonic data can also be provided, but we consider the fluorescence readout to be the most reliable way to verify the trapping. (The

plasmonic signal is more susceptible to drift and disturbances during the measurements, but excellent for verifying the gate operation.)

Figure R1:

Regarding the fluorescence data from enzymatic reactions with HRP and Amplex red, figure 6d is already showing two different experiments (time trace in one case, end values in the other). Additional data is available here as well, but the variation is high due to the chemical instability of the dye. The contrast (as compared to the absence of lactose) can be excellent, as shown by the example in Figure R2. However, presenting this result in the main text would be cherry picking, since some fluorescence from Amplex red was normally observed even without the cascade reaction running.

Figure R2:

Throughout the manuscript, several clarifications have been added on what data distributions show and experimental reproducibility.

Suggested Improvements: See the list of my comments below. In addition to these, the authors could try another enzyme cascade for further proof of concept. The cascade they have used is an excellent choice since it is well characterised in literature, and it nicely shows the entry of substrate and the exit of product. Another idea might be to use an enzyme or enzyme cascade that uses a cofactor such as nicotinamide, the reduced form of which, absorbs

at 340 nm and could be easily detected outside the chamber. This concept could be extended if an enzyme pair was used that recycled the cofactor back and forth between each other- enzyme 1 could be trapped in the chamber and enzyme 2 outside – recycling the cofactor back and forth through the permeable gateway would be a beautiful result.

We agree that the transport of cofactors through the polymer brush barrier is important to investigate. We have performed the requested experiments using alcohol dehydrogenase and nicotinamide (NAD⁺/NADH) as cofactor. We could confirm that cofactors were transported through the polymer brush by fluorescence (new figure S12). Furthermore, with the enzyme in the nanochambers, NAD⁺ could be reduced and transported out again as NADH, as detected by the absorbance increase. Additionally, again as suggested by the reviewer, we also showed that if enzymes which oxidize NADH are present outside the chambers (GOX + HRP), the cofactor is cycled back and forth (new figure S13).

In their conclusions, the authors suggest that their approach will enable the study of oligomerisation at high protein concentrations by FRET- but at the start of the paper in the introduction, they claim that its power lies in the potential to be used for analysing “single or small numbers of proteins” – again, as I have outlined for the abstract, I find this confusing- please clarify that it is both. Apart from oligomerisation, the authors should list the other possible ways in which their approach could be exploited since they write “...several studies relevant for molecular biology can be envisioned...”

One claim is that it could be used to study single protein molecules – what behaviour in a single protein molecule could be monitored and how? Would it be possible to detect enzyme motions in one molecule somehow?

Yes, this and many other things should be possible to study by established methods such as FRET. However, single proteins can be captured and studied also with vesicles (e.g. ref. 17). The main advantage of our platform is then the wider possibilities to easily introduce small ligands (or substrates) to the captured protein. We have added a few sentences in the final discussion to clarify this.

Clarity and Context and References: Please see comments regarding addition of extra references. Also, the paper mostly discusses the use of the approach to study proteins at very high concentrations but also mentions its potential for studying single protein molecules- the authors need to include more citations and discussion on the latter. And also, be clear in the distinction.

As mentioned above, the abstract and introduction has been clarified. Two references on single molecule monitoring are included.

Comments

1. The abstract opens with a sentence stating that the ability to detect and analyze single or few biological molecules has proven very important for understanding their function and interaction mechanisms, then a few sentences later the authors point out that it has been more difficult to entrap biomolecules under “physiological conditions”- I think the author needs to be more specific in what they refer to here by “physiological conditions” which I believe they mean to be high local concentrations of biomolecules entrapped inside cellular compartments.

This is a good point! Actually, here we meant that the proteins are not tethered and in a buffer with physiological salt content. However, the reviewer is right that the high concentration of other proteins is also an aspect of physiological conditions. It has been clarified in the discussion.

Then at the end of the abstract the authors claim that their approach will extend the possibility of single/few molecules analysis – I found this all to be muddled: first saying that this has already been tackled in literature then saying but it hasn't under physiological (high local concs and entrapped) but then back to single/few molecules. The abstract needs to be clarified in this respect. Are the authors interested in achieving highly concentrated/crowded conditions under nanoconfinement or are they interested in single/few entities being nanoconfined? Or both? I think it is both.

The reviewer is right that our platform is intended for both single and multiple protein entrapment. This comment is similar to previous ones and several clarifications have been made as already described.

2. The 3 statements in the bullet points below are quite sweeping – I think it is important that the authors cite more work in this field and have suggested citations below which show other work in which entrapment of proteins at high local concentration inside compartments has been achieved as has the ability for small molecules to cross the barrier. These references should be included.

Abstract: reaching concentrations up to 60 g/L, which is orders of magnitude higher than for existing container-based technologies.

Page 4: However, even the liposome trapping method has clear limitations. First, the yield is not very high, i.e., it is difficult to trap many proteins inside the same vesicle, which limits the possibilities to study interactions. Second, it is not straightforward for small molecules to cross the lipid membrane and access the entrapped protein, which limits studies of, for instance, ligand-induced conformational changes.

Page 5: The most important advantages of this technology in comparison with all existing trapping methods is that very large numbers of proteins can be confined very easily in an extremely small volume

The authors should point out other examples where entrapment at a high yield and also the ability for small molecules to cross the barrier has in fact been achieved, for example, this work by Bai and coworkers involving protein cages doi.org/10.1021/acscatal.1c02221 which entrapped two enzymes at 0.89 and 5 mM in a compartment with 65 nm diameter which was permeable to NADP(H).

We agree with the reviewer and we thank for pointing out the need to clarify how our system differs from those in the literature.

The work by Bai et al. (and others) uses fusion proteins constructed such that the self-assembling capsule is linked to the enzymes, which forces a large number of them to the interior wall of nanoscale carrier. However, these enzymes are not encapsulated in a tether-free manner, as they are in our nanochambers. This agrees well with the statement in the review by Kuchler et al. (ref. 64) which says “efficient entrapment of enzymes inside protein cages is yet to be achieved, unless the enzyme is part of the inner surface of the shell”. Thus, our work shows the first example of nanochambers with high concentration of proteins/enzymes that are not tethered. The free movement of enzymes (within the chambers) is likely a requirement for optimal activity as well as substrate channeling mechanisms.

Furthermore, our system is much more generic and compatible with activity monitoring at the level of individual nanochambers. In comparison, the fusion protein constructs diffuse around in solution and are analyzed at the ensemble level.

And also this work DOI: 10.1126/science.aaz6802 by Erb and coworkers involving the encapsulation of a 16-enzyme crotonyl-CoA carboxylase/reductase cascade

This work is interesting but uses microemulsions that are a billion times larger in volume than our chambers. Entire compartments from plants are encapsulated. Our work rather aims to construct organelles *bottom up*. We have cited the paper and mentioned this approach, but we consider it quite different.

Other work which could be cited includes this work by Cornelissen and coworkers DOI: 10.1039/c0sc00407c; Douglas and coworkers doi.org/10.1021/cb4006529;

We agree that both these papers are relevant and they have been cited. However, they again show fusion protein constructs where the enzymes are attached to the interior walls and not in solution phase.

Perhaps MOFS and COFS should also be discussed in the context of this work – for example, even the much-tested model system that the authors have used (glucose oxidase (GOx) and horseradish peroxidase(HRP)) as their proof of concept, has also been used in a MOF doi.org/10.1002/anie.201710418 this work might be worth discussing as a comparison.

We agree that this paper certainly has some relevance. However, it is highly focused on GOX and its immobilization on MOFs. Our work aims to avoid enzyme immobilization altogether and our cascade also includes galactosidase and lactose breakdown. Therefore, we do not see this paper as highly relevant considering that the literature has countless of studies on enzymes, especially GOX, operating *in vitro*. MOFs are anyhow mentioned in the new review paper we cite (ref. 68).

3. Page 7- a reference is needed for this sentence – “...gold film is introduced on top from which polymers are grafted by material-specific chemistry” what is the material-specific chemistry?

Thanks for pointing this out. The specificity to gold is obtained by thiol chemistry. We have clarified and cited appropriate references.

4. Page 10 Figure 2B: I found this figure confusing because at first glance, I mistakenly assumed that the two heating events (just after 0 min and at ~50 min) were aligned with the illustration above – in other words I thought the PNIPAM layer remained thick until the third heating event after 100 minutes (the red panel for this one being aligned to the illustration above in which the PNIPAM layer had collapsed). I may be being pedantic here, but I think that the authors should consider this accidental alignment. Also, explain the figure in more detail for the readers who are interested in enzyme entrapment but who are non-experts in plasmon resonance data.

We agree. We have added arrows in the figure to clarify that there are three heating events. The figure caption has been updated with a brief explanation.

5. Figure 4D: could temperature be added as another X axis at the top? This would show the precise temperature at ~100 s.

This comment must be referring to Figure 2D. Unfortunately, the temperature could not be measured in the custom liquid cell. It would anyway be very difficult to get an accurate reading exactly at the AFM tip. For this experiment we can only confirm that the heating was sufficient to reach above the LCST.

6. Figure S4A: the background signal is ~50 Hz and then BSA is injected which causes the signal to decrease to below 40 Hz then rinsing occurs at ~50 min after which, the signal remains unchanged. At 70 min I presume the system is cooled because the pale red shaded area returns to white (write heating above the red area as was done for figure 4A)- at this point the signal decreases further to ~ -18 Hz so, is this indicative of maximum BSA adsorption? I presume it is. Then when the pH is changed to 8 the signal increases which must be indicative of BSA desorbing- why then, does the signal not return to the original baseline if all the BSA has desorbed?

The reviewer is correct that a saturated layer of BSA corresponds to approximately 20 Hz. There is, however, always some experimental error due to baseline drift. Especially for measurements that last for several hours and where temperature is changed, a few Hz uncertainty is expected. The final value (~3 Hz) is most reasonably interpreted as noise in this case (especially since it is higher than the initial value). Since we have so many other control experiments showing BSA desorption (see Figures S5, S7 and S8 in SM) we are confident in this conclusion.

7. Figure S5: this is a control using PEG on gold which the authors state is protein-repelling- the maximum peak resonance shift is 3.5 (dip) and ~1.8 (peak)- the authors state that these values are not very different to those obtained in figure 3B which are 1.4 (dip) and 0.3 (peak). Are these considered similar for this technique?

This is probably a minor misunderstanding. In Figure S5, we also show the signals from the binding of the PEG layer. Therefore, the protein signal is smaller than the value on the y-axis. The signals to be compared differ by less than 0.2 nm in the end, which is comparable to the uncertainty from baseline drift. We have clarified the caption.

8. Page 16: the description of the experiment in Figure 4A: I recommend a more detailed description of the time course in 4A as it is the main experiment which proves the technique. For instance, there is a second injection of BSA just after 100 secs. Why was this required since BSA had already been injected at the start of the experiment? Was the BSA that had been injected first, washed away? Or did it remain in the surrounding solution ready to enter the pores? I assumed the latter but if it is the former then add more detail to the figure legend.

There are several injections and rinsing steps for the protein. We have changed the figure and the caption. We introduced more drawings to show what happens when along the time x-axis.

9. Figure S6: QCM data which is presented as frequency shift and dissipation change – For convenience, the authors should present the data as weight change too- this would have helped with understanding of figure S4A also.

We prefer to be careful with such quantification because, in general, the QCM response is not easy to translate into surface coverage. It is possible to use the Sauerbrey constant, but it highly overestimates the mass due to the coupled solvent. Here it gives $<100 \text{ ng/cm}^2$ for the $\sim 5 \text{ Hz}$ signal. We have explained this in the caption.

10. Supplementary note 4: Last two sentences – suggestion of imidazole as a way to desorb the enzymes would not be ideal since the solution in the chamber would then have a very high ionic strength which would not be considered ideally physiological – a selling point of this work.

This is a good point. However, the ionic strength can probably be reduced again after the desorption step, when all divalent ions have imidazole bound instead. We have revised the text.

11. Page 22 This sentence is sweeping, “...60 g/L. Notably, this is orders of magnitude higher than existing container-based trapping methods.” The authors should cite the work I have listed in point number 2 above which show other entrapment methods that achieve this – another notable approach that should be cited, because it achieved extremely high concentrations of proteins trapped in a porous electrode which is also accessible to small molecules and ions, can be found here DOI:10.1021/acs.chemrev.2c00397 or here <https://doi.org/10.1073/pnas.221412312>

We agree that the sentence should be revised. We have clarified that it refers to tether-free trapping. We also agree that the review paper is relevant and it has been cited. In particular, the electrochemical control should be possible to implement in our system which has a continuous metal film.

REVIEWER COMMENTS

Reviewer #1 (Remarks to the Author):

The authors have addressed all the concerns raised during the first round reviewing. I have no more technical questions. However, the authors mentioned a couple of times in the manuscript (page 23, 28) that their device could be used to trap single protein. I wonder how this can be controlled to achieve a single capture. If this is done by dilution, many of the nanochambers would be empty.

Reviewer #2 (Remarks to the Author):

Overall, I am satisfied with the revisions made to the manuscript in response to my original comments.

I am pleased with the additional experiment performed, based on my suggestion: "This concept could be extended if an enzyme pair was used that recycled the cofactor back and forth between each other- enzyme 1 could be trapped in the chamber and enzyme 2 outside – recycling the cofactor back and forth through the permeable gateway would be a beautiful result." BUTY I have concerns about the experiment and the conclusion at the end of the results section in the manuscript which I now address:

My understanding of the experiment is as follows: in figure S12 it was verified that nicotinamide could indeed pass through the PNIPAM brush. Figure S13 then shows the experiment in which the authors have chosen an alcohol dehydrogenase (ADH) inside the chamber which generates NADH as it oxidises ethanol to acetaldehyde; the NADH then exits the chamber through the brush gate where it is used in the bulk solution by GOX in its catalysis of the reduction of glucose with the concomitant regeneration of NAD⁺ from the NADH that exited the chamber (produced by the ADH reaction inside); this NAD⁺ can then enter the chamber again to be used by the ADH inside again, and so the back and forth recycling of the nicotinamide cofactor proceeds. The authors have decided to include ferrocyanide and horseradish peroxidase in the system (written in the figure S13 legend) , which they claim were added as "...additional redox mediators..." and this is very confusing because there is absolutely no requirement for electron mediators in this system because all redox reactions, for example the interconversion of NAD⁺/NADH, are carried out by each

enzyme, i.e. this is a simple hydrogen borrowing scheme and so I do not understand why electron mediators were added and it is very misleading to have them in there.

In addition to this, in the main manuscript on page 28 the last sentence before the Discussion section states: “It should even be possible to use the continuous metal film to implement electrochemical control of cofactor regeneration.⁶⁸” This is not correct. Simply having a conductive surface (gold) involved, does not mean that by applying a voltage to this gold that it would then be possible to control cofactor regeneration electrochemically; far from it, in fact. The direct (i.e. non-enzymatic) electrocatalysis of the reduction of NAD⁺ and the oxidation of NADH, requires a massive overpotential and even more importantly, without an enzyme, it is not stereoselective and would not have 100% fidelity for transfer at the nicotinamide’s C4 position as the proton can attack C6 instead OR two NADP⁺ radicals can combine to form a dimer.

The authors then cite ref 68 at the end of their sentence – this referenced paper 68 is about a novel electrochemical NAD(P)(H) recycling technology which DOES carry out the stereo-specific interconversion of NADP⁺/NADPH electrochemically, by exploiting an enzyme central to photosynthesis that can exchange electrons directly with the conductive electrode surface and then use them to catalyse the interconversion of NADP⁺/NADPH – in other words, it is an enzymatic interconversion which is driven electrochemically – a very special and unusual case made possible by the transduction of electricity into mobile electrons in the form of hydride on NADPH. In my first bout of comments, I highlighted this as a reference that the authors should cite, not because of its electrochemical nature, but because it achieves massively high concentrations of enzymes entrapped TETHER-FREE inside pores- please see my original comment pasted below:

11. Page 22 This sentence is sweeping, “....60 g/L. Notably, this is orders of magnitude higher than existing container-based trapping methods.” The authors should cite the work I have listed in point number 2 above which show other entrapment methods that achieve this – another notable approach that should be cited, because it achieved extremely high concentrations of proteins trapped in a porous electrode which is also accessible to small molecules and ions, can be found here DOI:10.1021/acs.chemrev.2c00397 or here <https://doi.org/10.1073/pnas.221412312>

Therefore I think the authors should not suggest that they can convert their system to an electrochemical one just because gold is involved.

Reviewer #1:

The authors have addressed all the concerns raised during the first round reviewing. I have no more technical questions. However, the authors mentioned a couple of times in the manuscript (page 23, 28) that their device could be used to trap single protein. I wonder how this can be controlled to achieve a single capture. If this is done by dilution, many of the nanochambers would be empty.

We thank the reviewer for the positive feedback. Indeed, while our technology is mostly intended for trapping multiple proteins per nanochamber, it could also be used for capturing single molecules as explained in the main text. It is correct that by chance, many chambers will then be empty, just like when lipid vesicles or other carriers are used for random encapsulation. This is counteracted simply by having many chambers and collecting data only from those that provide a fluorescent signal. Again, the strategy is the same when lipid vesicles are used. The advantage with our platform is that introduction of small molecules to the trapped (single) proteins becomes easier through the brush barrier. We have added one clarifying sentence in the main text.

Reviewer #2:

Overall, I am satisfied with the revisions made to the manuscript in response to my original comments. I am pleased with the additional experiment performed, based on my suggestion: “This concept could be extended if an enzyme pair was used that recycled the cofactor back and forth between each other- enzyme 1 could be trapped in the chamber and enzyme 2 outside – recycling the cofactor back and forth through the permeable gateway would be a beautiful result.”

We thank the reviewer for the positive feedback. It is nice to hear that the new experiments were appreciated.

BUTY I have concerns about the experiment and the conclusion at the end of the results section in the manuscript which I now address: My understanding of the experiment is as follows: in figure S12 it was verified that nicotinamide could indeed pass through the PNIPAM brush.

This is correct.

Figure S13 then shows the experiment in which the authors have chosen an alcohol dehydrogenase (ADH) inside the chamber which generates NADH as it oxidises ethanol to acetaldehyde; the NADH then exits the chamber through the brush gate where it is used in the bulk solution by GOX in its catalysis of the reduction of glucose with the concomitant regeneration of NAD⁺ from the NADH that exited the chamber (produced by the ADH reaction inside); this

NAD⁺ can then enter the chamber again to be used by the ADH inside again, and so the back and forth recycling of the nicotinamide cofactor proceeds.

Yes, these are the main reactions. The NADH cycling is essentially the same as in Lv et al. *Chem. Commun.* 2020, 56, 2723-2726 (new ref. 59). However, the “concomitant regeneration” of NAD⁺ from NADH is not spontaneous. GOX simply produces H₂O₂ and does not directly oxidize NADH. (In contrast, ADH does directly reduce NAD⁺.) Indeed, Lv et al. used special microparticles with peroxidase-like activity to complete the cycle via the generation of hydroxyl radicals. With only GOX and glucose, NADH remains reduced (at least over the timescale of the experiments). This is in agreement with our results (Figure R3). We could see the absorbance from NADH but it did not decrease over time. Direct addition of H₂O₂ to a solution with NADH did not lead to a reduction in absorbance either. This confirms that additional species are needed to regenerate NAD⁺. (All reactions were verified in bulk solution in this manner before tested in the nanochambers.)

Figure R3:

The authors have decided to include ferrocyanide and horseradish peroxidase in the system (written in the figure S13 legend) , which they claim were added as “...additional redox mediators...” and this is very confusing because there is absolutely no requirement for electron mediators in this system because all redox reactions, for example the interconversion of NAD⁺/NADH, are carried out by each enzyme, i.e. this is a simple hydrogen borrowing scheme and so I do not understand why electron mediators were added and it is very misleading to have them in there.

As shown above, additional catalysts are needed to make H₂O₂ oxidize NADH to NAD⁺. Lv et al. found that their microparticles had similar catalytic activity as HRP, which is why we included this enzyme. However, NADH is not a

cofactor/substrate for HRP, which is why ferricyanide was also included to make the NAD^+ generation more efficient. It is well established that ferricyanide easily participates in redox reactions with many species, including NADH and HRP. To illustrate its effect, we have included another experiment in Figure S13 which shows cofactor cycling with HRP included but without ferricyanide, which leads to a clear absorbance increase, i.e. the cycling is less efficient. Additionally, the figure legend has been rewritten to explain all of the above. We also realize that our initial reaction scheme was not very clear, so we have made a new figure to present it (Figure S14).

In addition to this, in the main manuscript on page 28 the last sentence before the Discussion section states: “It should even be possible to use the continuous metal film to implement electrochemical control of cofactor regeneration.⁶⁸” This is not correct. Simply having a conductive surface (gold) involved, does not mean that by applying a voltage to this gold that it would then be possible to control cofactor regeneration electrochemically; far from it, in fact. The direct (i.e. non-enzymatic) electrocatalysis of the reduction of NAD^+ and the oxidation of NADH, requires a massive overpotential and even more importantly, without an enzyme, it is not stereoselective and would not have 100% fidelity for transfer at the nicotinamide’s C4 position as the proton can attack C6 instead OR two NADP^+ radicals can combine to form a dimer. The authors then cite ref 68 at the end of their sentence – this referenced paper 68 is about a novel electrochemical NAD(P)(H) recycling technology which DOES carry out the stereo-specific interconversion of $\text{NADP}^+/\text{NADPH}$ electrochemically, by exploiting an enzyme central to photosynthesis that can exchange electrons directly with the conductive electrode surface and then use them to catalyse the interconversion of $\text{NADP}^+/\text{NADPH}$ – in other words, it is an enzymatic interconversion which is driven electrochemically – a very special and unusual case made possible by the transduction of electricity into mobile electrons in the form of hydride on NADPH. In my first bout of comments, I highlighted this as a reference that the authors should cite, not because of its electrochemical nature, but because it achieves massively high concentrations of enzymes entrapped TETHER-FREE inside pores- please see my original comment pasted below: 11. Page 22 This sentence is sweeping, “...60 g/L. Notably, this is orders of magnitude higher than existing container-based trapping methods.” The authors should cite the work I have listed in point number 2 above which show other entrapment methods that achieve this – another notable approach that should be cited, because it achieved extremely high concentrations of proteins trapped in a porous electrode which is also accessible to small molecules and ions, can be found here DOI:10.1021/acs.chemrev.2c00397 or here <https://doi.org/10.1073/pnas.221412312> Therefore I think the authors

should not suggest that they can convert their system to an electrochemical one just because gold is involved.

Thank you for this comment! The reviewer is correct that electrochemical control does not directly enable cofactor regeneration. To avoid confusion we have simply removed this statement. We also understand now that the reviewer did not mention the articles about the “leaf” because of the electrochemical aspects. We have cited it in relation to its high enzyme immobilization capacity and novel applications instead.

While it is technically correct that the enzymes in the nanoporous electrodes are immobilized tether-free in the sense that there is no chemical tether attached to them, they are nevertheless in contact with a surface and not free in solution phase, as far as we understand. (Otherwise they would not accumulate in the porous material and/or they would leak out.) That being said, we have pointed out that catalytic activity may sometimes be well preserved even if the enzymes are not free in solution-phase.

This discussion made us realize that we have not use the terms “tether-free” and “solution-phase” in an ideal way, so we made some clarifications. For instance, various nanocarriers can be said to be in solution-phase even though the enzymes inside are attached to the walls. The advantage of our system is that the enzymes themselves are in solution phase (and tether free). This is a unique feature, but it does not mean that our system is superior to existing technologies in all cases.

REVIEWERS' COMMENTS

Reviewer #2 (Remarks to the Author)

1. First of all – my apologies for not looking in more detail at the enzyme pair chosen by the authors for my suggested experiment; I envisaged the experiment as a simple hydrogen borrowing cascade i.e. an enzyme pair in which one of the enzymes produces NADH which is subsequently used by enzyme number 2 which then produces NAD⁺ again to be used by enzyme 1, and so I assumed that was what the authors had done – but in fact they chose a more complex system using an alcohol dehydrogenase (uses NAD⁺) and glucose oxidase (which does not use nicotinamide at all – I should have realised this second enzyme, GOX, did not require nicotinamide- with hindsight it is obvious to me as it is a common enzyme – apologies).

Now with fresh eyes, I understand the experiment to be based on the paper by Lv et al who also use ADH and GOX along with a special particle which converts H₂O₂ to a hydroxyl free radical – this radical then “initiates” the oxidation of NADH to NAD⁺ thus regenerating it for the ADH to use.

The experiment here was inspired by Lv et al’s paper and instead of the special particle, the authors have included HRP.

As a control, the authors tested the ability of H₂O₂ alone to oxidise the NADH back to NAD⁺ but it did not.

In their answers the authors write “However, NADH is not a cofactor/substrate for HRP, which is why ferricyanide was also included to make the NAD⁺ generation more efficient.”

Here are a few papers which show that HRP in fact can oxidise NADH, so I am a bit confused as to why ferri/ferro was needed, however in the legend of S13 the authors write “The effect from HRP without ferricyanide was smaller.” so perhaps they need to be less definite about stating that HRP does not oxidise NADH, perhaps it does since they see an effect, albeit smaller as they point out.

<https://doi.org/10.1021/ja0585735>

<https://doi.org/10.1093/jb/mvs068>

<https://doi.org/10.1111/j.1751-1097.1978.tb07013.x>

But I may be missing something here and I don’t want to hold up the publishing of this work

further, so perhaps if the authors address this and maybe add a citation where ferri/ferro has been used before with HRP, then that's all good and I do not need to review this again. e.g. <https://pubs.acs.org/doi/pdf/10.1021/bi00827a015>

and

The oxidation-reduction potentials of compound I/compound II and compound II/ferric couples of horseradish peroxidases A2 and C. - ScienceDirect

2. In response to the authors understanding of the references on the "Leaf" – in fact in this system only one of the enzymes is in direct contact with the inner surface of the pores by being adsorbed to it – the rest of the enzymes and enzyme cascades are not required to make electrical contact with the pore inner wall and can be free in solution in the lumen of the pores but this is not specifically stated/proved and it is likely that whilst some are "free" in the solution in the pore lumen, some are also not, so no further changes regarding this point are required.

Reviewer #2:

1. First of all – my apologies for not looking in more detail at the enzyme pair chosen by the authors for my suggested experiment; I envisaged the experiment as a simple hydrogen borrowing cascade i.e. an enzyme pair in which one of the enzymes produces NADH which is subsequently used by enzyme number 2 which then produces NAD⁺ again to be used by enzyme 1, and so I assumed that was what the authors had done – but in fact they chose a more complex system using an alcohol dehydrogenase (uses NAD⁺) and glucose oxidase (which does not use nicotinamide at all – I should have realised this second enzyme, GOX, did not require nicotinamide- with hindsight it is obvious to me as it is a common enzyme – apologies).

No problem! The reviewer has helped us improve this work by providing very constructive criticism. It is true that there are probably other enzyme pairs that would not require HRP, but for our model experiment we chose to simply follow the protocol in Lv et al.

Now with fresh eyes, I understand the experiment to be based on the paper by Lv et al who also use ADH and GOX along with a special particle which converts H₂O₂ to a hydroxyl free radical – this radical then “initiates” the oxidation of NADH to NAD⁺ thus regenerating it for the ADH to use. The experiment here was inspired by Lv et al’s paper and instead of the special particle, the authors have included HRP. As a control, the authors tested the ability of H₂O₂ alone to oxidise the NADH back to NAD⁺ but it did not.

This is correct.

In their answers the authors write “However, NADH is not a cofactor/substrate for HRP, which is why ferricyanide was also included to make the NAD⁺ generation more efficient.”

Here are a few papers which show that HRP in fact can oxidise NADH, so I am a bit confused as to why ferri/ferro was needed, however in the legend of S13 the authors write “The effect from HRP without ferricyanide was smaller.” so perhaps they need to be less definite about stating that HRP does not oxidise NADH, perhaps it does since they see an effect, albeit smaller as they point out.

<https://doi.org/10.1021/ja0585735>

<https://doi.org/10.1093/jb/mvs068>

<https://doi.org/10.1111/j.1751-1097.1978.tb07013.x>

The reviewer is right. We have refrained from writing that HRP cannot react with NADH in the manuscript. The figure legend only states that the effect is smaller. One of the above references has been cited.

But I may be missing something here and I don't want to hold up the publishing of this work further, so perhaps if the authors address this and maybe add a citation where ferri/ferro has been used before with HRP, then that's all good and I do not need to review this again.

e.g. <https://pubs.acs.org/doi/pdf/10.1021/bi00827a015>

and

The oxidation-reduction potentials of compound I/compound II and compound II/ferric couples of horseradish peroxidases A2 and C. - ScienceDirect

One of the above references has been cited. We also added a reference about the reaction between ferricyanide and NADH:

Komor, E., Thom, M. & Maretzki, A. The oxidation of extracellular NADH by sugarcane cells: Coupling to ferricyanide reduction, oxygen uptake and pH change. *Planta* 170, 34-43 (1987)

2. In response to the authors understanding of the references on the "Leaf" – in fact in this system only one of the enzymes is in direct contact with the inner surface of the pores by being adsorbed to it – the rest of the enzymes and enzyme cascades are not required to make electrical contact with the pore inner wall and can be free in solution in the lumen of the pores but this is not specifically stated/proved and it is likely that whilst some are "free" in the solution in the pore lumen, some are also not, so no further changes regarding this point are required.

OK